# *Scn1a* gene reactivation after symptom onset rescues pathological phenotypes in a mouse model of Dravet syndrome

Nicholas Valassina[1,8], Simone Brusco[1,8], Alessia Salamone[1], Linda Serra[1], Mirko Luoni[1], Serena Giannelli[1], Simone Bido [1], Luca Massimino[2], Federica Ungaro [2], Pietro Giuseppe Mazzara [1,7], Patrizia D'Adamo [3,4], Gabriele Lignani [5], Vania Broccoli [1,6,9 ✉] & Gaia Colasante [1,9 ✉]

Dravet syndrome is a severe epileptic encephalopathy caused primarily by haploinsufficiency of the *SCN1A* gene. Repetitive seizures can lead to endurable and untreatable neurological deficits. Whether this severe pathology is reversible after symptom onset remains unknown. To address this question, we generated a *Scn1a* conditional knock-in mouse model (*Scn1a* Stop/+) in which *Scn1a* expression can be re-activated on-demand during the mouse lifetime. *Scn1a* gene disruption leads to the development of seizures, often associated with sudden unexpected death in epilepsy (SUDEP) and behavioral alterations including hyperactivity, social interaction deficits and cognitive impairment starting from the second/third week of age. However, we showed that *Scn1a* gene re-activation when symptoms were already manifested (P30) led to a complete rescue of both spontaneous and thermic inducible seizures, marked amelioration of behavioral abnormalities and normalization of hippocampal fast-spiking interneuron firing. We also identified dramatic gene expression alterations, including those associated with astrogliosis in Dravet syndrome mice, that, accordingly, were rescued by *Scn1a* gene expression normalization at P30. Interestingly, regaining of $Na_v1.1$ physiological level rescued seizures also in adult Dravet syndrome mice (P90) after months of repetitive attacks. Overall, these findings represent a solid proof-of-concept highlighting that disease phenotype reversibility can be achieved when *Scn1a* gene activity is efficiently reconstituted in brain cells.

[1] Stem Cell and Neurogenesis Unit, Division of Neuroscience, IRCCS San Raffaele Scientific Institute, 20132 Milan, Italy. [2] Humanitas Clinical and Research Center - IRCCS, Rozzano, Milan, Italy. [3] Molecular Genetics of Intellectual Disability, IRCCS San Raffaele Scientific Institute, 20132 Milan, Italy. [4] Mouse behavior Core Facility, IRCCS San Raffaele Scientific Institute, 20132 Milan, Italy. [5] Department of Clinical and Experimental Epilepsy, UCL Queen Square Institute of Neurology, London, UK. [6] National Research Council (CNR), Institute of Neuroscience, 20129 Milan, Italy. [7] Present address: Department of Genetics and Development, Columbia University, 10032 New York, NY, USA. [8] These authors contributed equally: Nicholas Valassina, Simone Brusco. [9] These authors jointly supervised this work: Vania Broccoli, Gaia Colasante. ✉email: broccoli.vania@hsr.it; colasante.gaia@hsr.it

Dravet syndrome (DS) is a devastating epileptic encephalopathy arising in otherwise normal babies in the first year of life, later accompanied by developmental delay, intellectual disability and mood disorders. In the 80% of cases, it is caused by haploinsufficiency of *SCN1A* gene, that encodes for the alpha subunit of the voltage-gated sodium channel (VGSC) $Na_v1.1$[1,2]. Current pharmacological treatments for DS are ineffective in completely control convulsive attacks or delaying subsequent neurological symptoms, and, thus, a number of gene-based therapeutic strategies are in development. Gene supplementation therapy is hardly feasible in DS, as *SCN1A* coding sequence largely exceeds the packaging cargo of adeno-associated viral vectors (AAV), that are commonly employed for therapeutic gene delivery in the CNS[3,4]. In recent years, alternative genetic approaches aiming to restore physiological levels of $Na_v1.1$ to treat DS have been developed. Those strategies rely on boosting the expression of the healthy copy of *SCN1A* gene at transcriptional[5–7] or post-transcriptional level[8,9] to rescue channel haploinsufficiency.

All these studies are supported by the assumption that DS pathological signs are reversible, but the possibility and the extent of symptom reversion after their full appearance is untested up to now.

In fact, the new therapeutic approaches have been delivered perinatally[5,6,9] or at some later stages (P15-P30)[7,9] and in the latter cases mild phenotypic amelioration has been achieved, thus suggesting that either the efficiency in increasing *Scn1a* gene expression is not sufficient for symptomatic recovery or that the pathology cannot be reverted anymore after symptoms have already manifested.

On this line, recent findings have shown that parvalbumin (PV) interneurons in DS exhibit a transient functional impairment in a specific time window between the second and third postnatal weeks (P11-21), coincident with *Scn1a* expression onset[10]. After P30, a normalization of PV interneuron excitability is reported suggesting that their action potential (AP) failure contributes to the appearance of epilepsy but is not the main responsible for the long-term chronic epilepsy in DS. Therefore, other yet unknown deficits in PV or other interneuron subtypes may contribute to the phenotype. In addition, compensatory reconfiguration of cortical circuits occurring in the first post-natal period in DS might also contribute to sustain chronic epilepsy in the stabilized phase of the syndrome. In this scenario, the assumption that DS symptoms can be reverted upon re-expression of the *Scn1a* gene at physiological levels requires a solid experimental validation.

In the present work, we conclusively addressed the symptomatic reversibility in DS by generating a mouse model of DS, in which one allele of *Scn1a* gene is silent (*Scn1a*[Stop/+]), and can be reconstituted in a functional allele upon Cre recombinase activity.

Normalization of *Scn1a* expression levels after symptom appearance was achieved by systemic delivery of an AAV expressing Cre recombinase in *Scn1a*[Stop/+] mice at P30 and it was sufficient to rescue seizures and behavioral alterations characteristic of this model and associated changes in gene expression. Interestingly, a complete recovery of the epileptic phenotype was achieved even when $Na_v1.1$ levels were normalized in adult DS mice (P90). Overall, these findings corroborate the concept that, even at later stages, optimal treatments can substantially recover DS epileptic and behavioral symptoms.

## Results

### Development and characterization of the *Scn1a*[Stop/+] mouse model.
To investigate the possibility to revert DS pathology upon restoration of $Na_v1.1$ physiological levels once that symptoms have already manifested, we generated a knock-in mouse model in which one *Scn1a* allele is inactivated but its expression can be conditionally restored. To disrupt *Scn1a* gene, we inserted a STOP cassette arrayed between loxP sites in the intron between exons 6 and 7 of the gene (Fig. 1a). The correct integration in the *Scn1a* gene locus was assessed by Southern blot analysis (Supplementary Fig. 1a, b). The STOP cassette can be conditionally removed upon Cre recombinase expression to reconstitute the functional *Scn1a* allele (Fig. 1a), thus enabling the conditional manipulation of the endogenous *Scn1a* genomic locus.

To assess the ability of the inserted STOP cassette in silencing gene expression and its correct removal by Cre mediated loxP sequence recombination, we crossed *Scn1a*[Stop/+] mice (129 Sv background) with CMV-Cre mouse strain (C57Bl6/N background) and analyzed the *Scn1a*[+/+], *Scn1a*[Stop/+] and *Scn1a*[Stop/+];*CMV-Cre* (hereafter as *Scn1a*[Rec/+]) mice. Genotype analysis correctly identified the STOP flox (535 bp amplicon) and the recombined (379 bp amplicon) alleles with the latter differing from the wild-type (345 bp amplicon) sequence for the presence of one residual loxP site (Fig. 1a, b). While no difference among genotypes was observed in the weight gain in the first 6 weeks of life (Fig. 1c), *Scn1a*[Stop/+] mice started to die of SUDEP similarly to previously described DS murine models[11–14], while *Scn1*[Rec/+] did not (Fig. 1d). *Scn1a*[Stop/+] mice were backcrossed to generate a *Scn1a*[Stop/Stop] progeny that showed a significant decrease in the weight and died within the second week of age (Fig. 1c, d)[12,14].

Quantitative PCRs (qPCR) performed with primers spanning either exon-1 or exon-7 on RNA from *Scn1a*[Stop/+] cerebral cortices and relative controls *Scn1a*[+/+] at 6 weeks showed a non-significant trend toward a 50% reduction of *Scn1a* gene expression at RNA level, that was normalized in *Scn1a*[Rec/+] (Fig. 1e). Accordingly, the levels of $Na_v1.1$ protein were halved in *Scn1a*[Stop/+] and completely recovered in *Scn1a*[Rec/+] (Fig. 1f, g).

Overall, these data confirmed that the STOP cassette properly inactivated the *Scn1a* knocked-in allele and that its Cre-mediated removal was able to restore normal protein levels of the channel.

Therefore, this mouse model offers the possibility to investigate DS reversibility after symptom onset.

### Perinatal normalization of $Na_v1.1$ level prevents SUDEP and seizures in DS model.
To achieve temporal control of *Scn1a* expression and restore physiological expression of $Na_v1.1$ after disease manifestation, we selected the AAV-PHP.eB[15], a synthetic AAV9 capsid capable of crossing the blood-brain barrier (BBB) after intravascular delivery, to express the Cre recombinase efficiently in both neuronal and glial cells throughout the brain. Delivery of PHP.eB-Cre ($3 \times 10^{11}$ viral genomes (vg)/mouse) by temporal vein injection in perinatal (P1-P2) *Scn1a*[Stop/+] (F1:129 Sv *Scn1a*[Stop/+] × C57BL/6 J) pups (Fig. 2a) prevented SUDEP in 90% of *Scn1a*[Stop/+] mice up to 120 days (4 months) after injection (Fig. 2b). To mimic febrile seizures, *Scn1a*[Stop/+] mice injected with either PHP.eB-Cre or PHP.eB-geneless (control, ctrl) were subjected to thermal induction protocol at 4 months of age. When challenged with gradual increase in body temperature until 42.5 °C, none of the PHP.eB-Cre injected *Scn1a*[Stop/+] mice displayed behavioral seizures, while all control *Scn1a*[Stop/+] mice experienced a seizure between 39 and 42.3 °C (Fig. 2c). Viral copy number analysis evidenced a prevalent targeting of the PHP.eB-Cre in the brain respect to the liver (Fig. 2d) and Western blot analysis highlighted that viral biodistribution in the brain was sufficient to regain normal levels of $Na_v1.1$ protein in cerebral cortex, hippocampus, cerebellum and striatum (Fig. 2e, f).

These data indicate that perinatal normalization of $Na_v1.1$ expression in a DS model prevents symptom onset, including

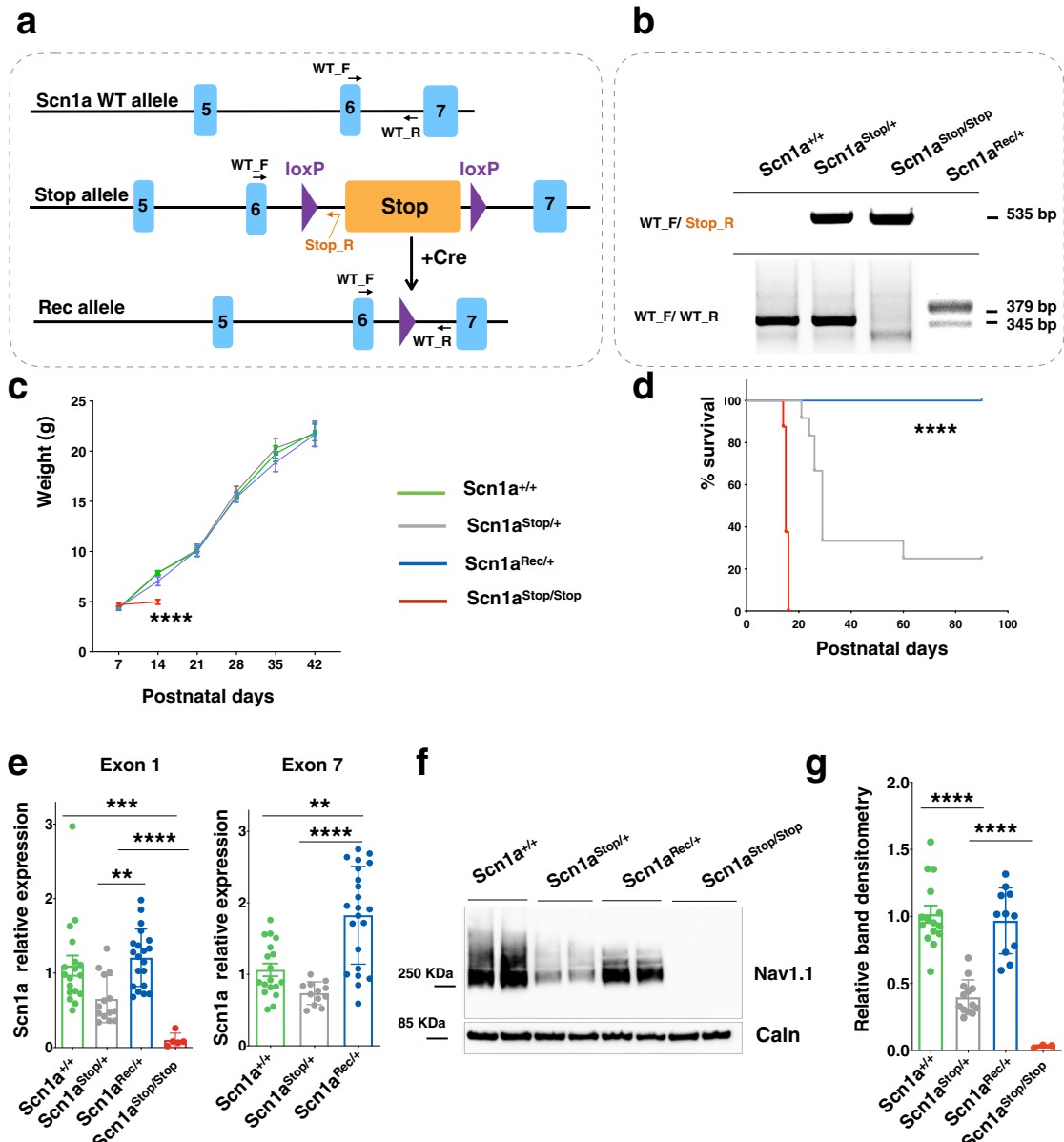

**Fig. 1 Characterization of a *Scn1a* knock-in mice carrying a floxed STOP cassette between exon 6 and exon 7. a** Schematic representation of the wild-type *Scn1a* allele, the knock-in allele containing the STOP cassette and the recombined allele after Cre-mediated removal of the STOP cassette. Grey boxes represent *Scn1a* exons and violet triangles LoxP sites. Primers for genotyping PCR are shown. **b** Genomic PCR products for genotyping *Scn1a*+/+, *Scn1a*Stop/+, *Scn1a*Stop/Stop, and *Scn1a*Rec/+ **c** Body weight curves (from P7 to P42) of *Scn1a*+/+, *Scn1a*Stop/+, and *Scn1a*Rec/+ and *Scn1a*Stop/Stop, mice (n = 8 each group, two-way ANOVA with Tukey's multiple comparison ****$p < 0.0001$ for genotype effect). **d** Survival of *Scn1a*Stop/+ (n = 12), *Scn1a*Rec/+ (n = 11) and *Scn1a*Stop/Stop (n = 8) mice monitored out to 90 days for survival, (****$p < 0.0001$; Log-rank Mantel-Cox test). **e** qRT-PCR on RNA extracted from cerebral cortex to determine *Scn1a* transcript level in *Scn1a*+/+ (n = 18), *Scn1a*Stop/+ (n = 14) and *Scn1a*Rec/+ n = 20) and *Scn1a*Stop/Stop (n = 6) mice with primers tagging exon 1 (**$p = 0.0026$; ***$p = 0.0007$; ****$p < 0.0001$) and exon 7 (**$p = 0.0044$; ****$p < 0.0001$, Kruskal-Wallis followed by Dunn's multiple comparisons test). **f** Western blot of membrane-enriched protein extracts prepared from cerebral cortex at 6 weeks of age (P14 for *Scn1a*Stop/Stop) to evaluate Na$_v$1.1 protein content. Calnexin was used as normalizer. **g** Densitometric quantification of Na$_v$1.1 immunoreactive bands normalized to Calnexin in western blots of *Scn1a*+/+ (n = 15), *Scn1a*Stop/+ (n = 13) and *Scn1a*Rec/+ (n = 11) and *Scn1a*Stop/Stop (n = 2) adult mouse cerebral cortices (****$p < 0.0001$ one-way ANOVA followed by Tukey's multiple comparison). Data are shown as mean ± SEM; where present, dots represent individual samples. Source data are provided as a Source Data file.

SUDEP and thermal-induced seizures and confirmed similar results obtained in recent studies[6,9].

**Normalization of Na$_v$1.1 levels after symptom onset rescues the seizure phenotype in DS model.** Next, we proceeded to assess the effect of Na$_v$1.1 level restoration after DS symptom onset. As symptoms start to appear between the second and the third week

of age, postnatal day 30 (P30) was chosen as a suitable time point for Na$_v$1.1 expression restoration (Fig. 3a). To determine the efficiency of viral brain transduction in adult mice, *Scn1a*Stop/+ animals injected either with control or Cre-expressing viruses at P30 ($10^{12}$ vg/mouse) were euthanized at P60 and brain tissues were dissected to extract RNA and membrane-enriched proteins (Supplementary Fig. 2a). Molecular analysis confirmed 50%

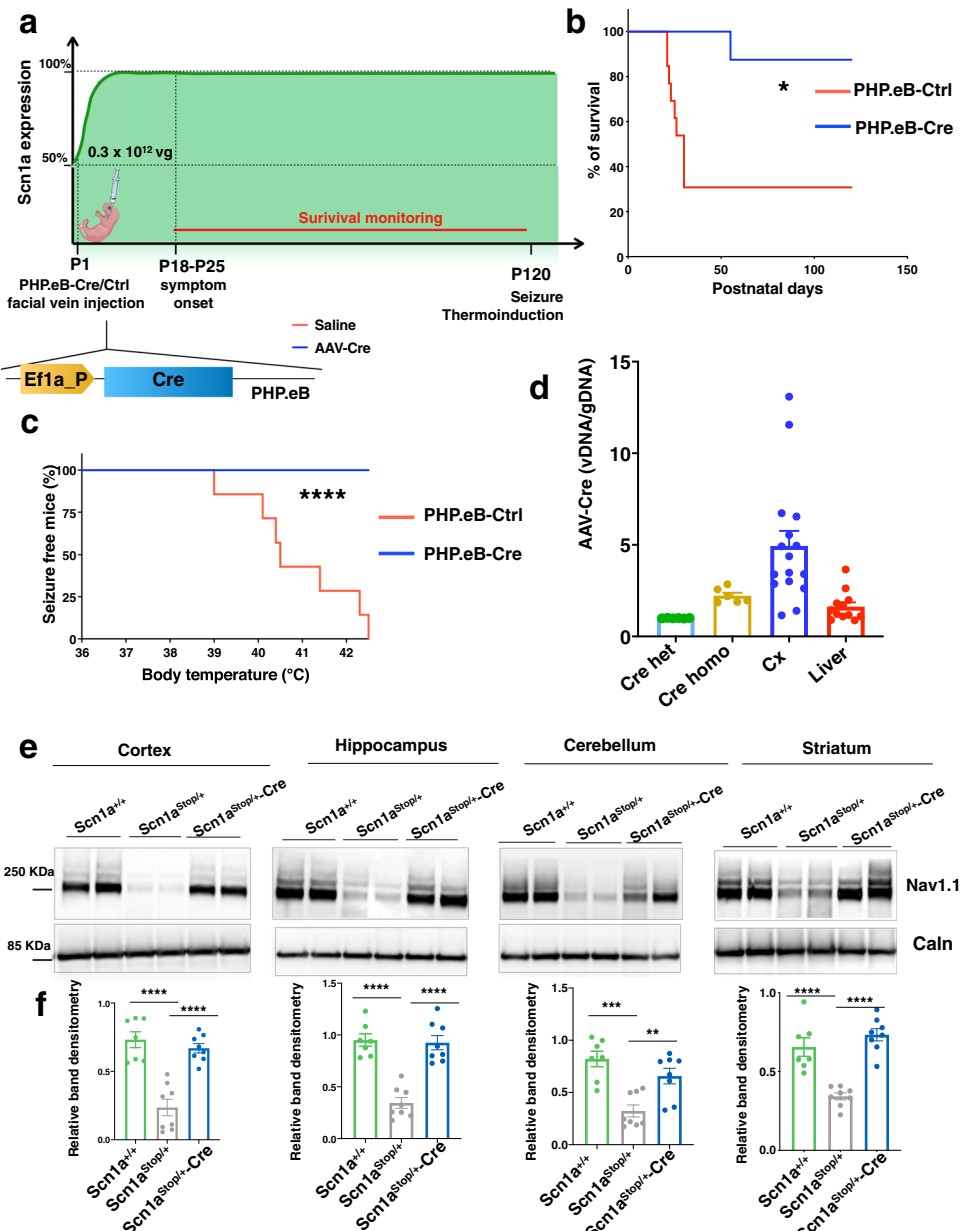

**Fig. 2 Perinatal normalization of Na$_v$1.1 level prevents SUDEP and seizures in DS model. a** Schematic representation of the experimental timeline, with *Scn1a*$^{Stop//+}$ mice undergoing PHP.eB-Ctrl or Cre facial vein injection at P1, followed by survival monitoring and seizure thermal induction at P120. Created with BioRender.com **b** Survival curve of *Scn1a*$^{Stop/+}$ -Ctrl (n = 13) and *Scn1a*$^{Stop//+}$ -Cre (n = 9) mice (*$p$ = 0.0108, Log-rank Mantel-Cox test). **c** Plot showing the percentage of seizure-free *Scn1a*$^{Stop/+}$ -Cre mice compared to *Scn1a*$^{Stop/+}$ -Ctrl mice (n = 7 for each group, ****$p$ < 0.0001; Mantel-Cox log-rank test). **d** Viral copy number analysis in mice injected with PHP.eB-Cre in the cerebral cortex and liver. **e, f** Western blot analyses of Na$_v$1.1 protein in the cortex, hippocampus, cerebellum and striatum in *Scn1a*$^{+/+}$, *Scn1a*$^{Stop/+}$-Ctrl and *Scn1a*$^{Stop/+}$-Cre mice and their relative band densitometry quantification normalized on calnexin, (****$p$ < 0.0001,***$p$ = 0.0002, **$p$ = 0.006, one-way ANOVA with Tukey's multiple comparison). Data are represented as mean ± SEM with dots representing individual samples. Source data are provided as a Source Data file.

reduction of *Scn1a* mRNA levels in cerebral cortex of *Scn1a*$^{Stop/+}$ mice compared to control *Scn1a*$^{+/+}$ and *Scn1a*$^{Stop/+}$-Cre mice (Supplementary Fig. 2b). Similar results were obtained analyzing Na$_v$1.1 protein level in the different brain compartments (Supplementary Fig. 2c–f). These findings validated the high efficiency of Cre-mediated *Scn1a* gene reactivation in all the analyzed brain regions achieved by viral systemic administration in P30 mice.

To further corroborate the wide viral brain transduction, we performed immunofluorescence for Cre protein on *Scn1a*$^{Stop/+}$ -Cre mice brain tissues (Supplementary Fig. 3a–a"). We observed that around 70% of cells were expressing Cre protein in the

cortex, striatum and dentate gyrus (DG) of hippocampus, while 85% in the cornu ammonis (CA) (Supplementary Fig. 3i). To better characterize transduction efficiency of neurons and glial cells of PHP.eB-AAV in the hippocampus, we injected PHP.eB-Cre in Ai14 mice and we assessed tdTomato reporter activation, observing that around 80% of total neurons and 20% of GFAP cells were transduced in this brain compartment (Supplementary Fig. 3b–e", j). Quantification of viral copy number in cerebral cortex and in the liver of the *Scn1a*$^{Stop/+}$-Cre mice showed a higher number of viral copies in the cerebral cortex with respect to the liver (Supplementary Fig. 3k) confirming that the PHP.eB

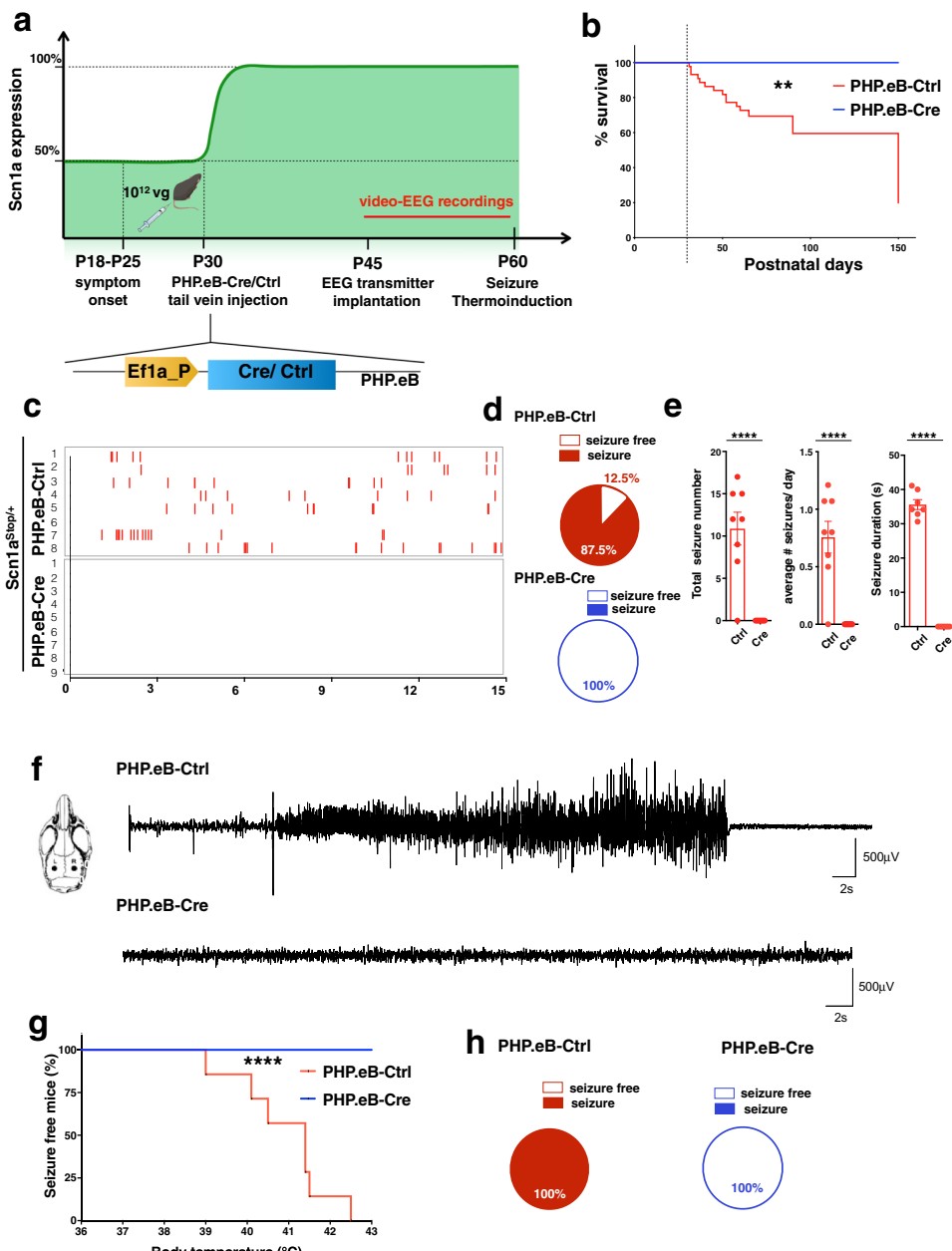

**Fig. 3 Re-expression of *Scn1a* gene after symptoms onset rescues seizures in DS mice. a** Schematics depicting the experimental timeline, with *Scn1a*Stop/+ mice undergoing PHP.eB-Ctrl or PHP.eB-Cre tail vein injection at P30, followed by EEG transmitter implantation at P45, two weeks of EEG recordings and, seizure thermal induction at P60. Created with BioRender.com. **b** Survival of *Scn1a*Stop/+ ($n = 23$), *Scn1a*Stop/+ -Cre ($n = 11$) monitored out to 90 days (**$p = 0.0018$; Log-rank Mantel-Cox test). **c** Raster plot showing all generalized tonic-clonic seizures (Racine scale stages 4 and 5) in 8 *Scn1a*Stop/+-Ctrl injected mice and 9 *Scn1a*Stop/+-Cre mice subjected to video-EEG recordings. **d** Pie chart (right panel) showing the proportion of Ctrl- and PHP.eB-Cre-injected *Scn1a*Stop/+ mice with or without spontaneous seizures ($p = 0.0004$; Fisher's exact test). **e** Cumulative seizure number (*left*), average daily seizure frequency in two weeks of recordings (*middle*) and average seizure duration (*right*), in *Scn1a*Stop/+ -Ctrl ($n = 8$) and -Cre ($n = 9$) injected mice. (****$p < 0.0001$, two-tailed unpaired test). Data are shown as mean ± SEM, with dots representing individual mice. **f** *Left*, schematic of electrode placement in left (L) and right (R) cortical hemispheres for EEG recordings (from bregma AP -1; ML ± 1). *Right*, Representative traces of EEG recordings of *Scn1a*Stop/+ -Ctrl injected during an ictal episode and *Scn1a*Stop/+ -Cre injected. **g** Percentage of *Scn1a*Stop/+ -Ctrl and *Scn1a*Stop/+ -Cre mice remaining seizure-free after thermal induction (*left*) ($n = 8$ *Scn1a*Stop/+-Ctrl and $n = 9$ for for *Scn1a*Stop/+ -Cre (****$p < 0.0001$; Mantel-Cox log-rank test). **h** Pie charts showing the percentage of *Scn1a*Stop/+-Ctrl and for *Scn1a*Stop/+ -Cre mice with or without induced thermal seizures ($p = 0.0004$; two-sided Fisher's exact test). Source data are provided as a Source Data file.

capsids have a robust efficiency to cross the BBB in adult mouse brains and to spread more efficiently across the neural tissue with respect to peripheral organs.

Once the viral efficiency for brain targeting was validated, we proceeded with the characterization of the functional effects

dependent on Na$_v$1.1 reactivation. *Scn1a*Stop/+ mice surviving the P18-P24 critical time window, but developing spontaneous seizures during routine handling, were randomized into two experimental groups and subjected to either control or Cre virus systemic administration (10$^{12}$ vg/mouse) (Fig. 3a). Survival curve

of Cre injected mice showed a complete rescue of SUDEP in the following 4 months in contrast with control treated $Scn1a^{Stop/+}$ mice, that continued to die (Fig. 3b). A subgroup of these animals, when they reached a suitable weight (~20 g, at P45), were implanted with wireless transmitters and EEG signals from somatosensory cortex (SSC) were recorded for 2 weeks coupled with video monitoring (video-EEG analysis). Tonic-clonic spontaneous seizures were observed in 7 out of 8 (87.5%) control treated $Scn1a^{Stop/+}$ mice (Fig. 3c, d, f), with a total number of around 10 seizures over the two weeks and a mean duration of 35 s (Fig. 3e). On the contrary, in all $Scn1a^{Stop/+}$ Cre mice (9 out of 9) continuous video/EEG monitoring did not show any spontaneous seizure (Fig. 3c–f), suggesting that $Scn1a$ gene reactivation was able to protect from behavioral and/or electrographic seizures. Then, a group of mice was subjected to hyperthermia induced seizures at P60. Interestingly, none of the $Scn1a^{Stop/+}$-Cre mice experienced seizures even when body temperature was raised to 43 °C, while all $Scn1a^{Stop/+}$-Ctrl mice developed tonic-clonic seizures (Fig. 3g, h). Around one hour after seizure thermo-induction, 2 out 9 $Scn1a^{Stop/+}$-Cre mice were found dead in their cages and subsequent Western blot analysis on their cerebral cortical and hippocampal tissues revealed that levels of $Na_v1.1$ were not fully re-established (Supplementary Fig. 4a, b), indicating that they received suboptimal viral transduction.

Altogether, these data show that re-expression of physiological levels of $Na_v1.1$ after symptom onset is sufficient to rescue spontaneous and thermo-induced seizures.

**Normalization of $Na_v1.1$ levels after symptom onset rescues behavioral alterations.** DS is accompanied by troublesome neuropsychiatric comorbidities, including hyperactivity, attention deficit, delayed psychomotor development, sleep disorder, anxiety-like behaviors, impaired social interactions, restricted interests, and severe cognitive deficits[16,17].

To profile behavioral alterations in our new DS model and assess their eventual recovery following $Na_v1.1$ protein re-expression, we subjected $Scn1a^{+/+}$, $Scn1a^{Stop/+}$ -Ctrl, and $Scn1a^{Stop/+}$-Cre male mice at 3–4 months of age to a comprehensive battery of tests (Supplementary Fig. 5a, b). We first assessed motor coordination and balance using an accelerating rotarod test. Mice were subjected to three trials a day for 5 consecutive days, and the latency to fall off from the rotatory rod was measured (Supplementary Fig. 5c). Although displaying a slightly inferior latency to fall compared with control $Scn1a^{+/+}$ mice, $Scn1a^{Stop/+}$ mice did not show significant motor coordination alterations over the course of the 5 days (Supplementary Fig. 5d), differently from other models previously described[11].

To investigate hyperactivity and anxiety behavior, the three experimental mouse groups were transferred from their familiar home cages to an open field arena (Fig. 4a). During a 10 min open-field test, adult $Scn1a^{Stop/+}$-Ctrl mice travelled significantly farther and faster than control mice indicating hyperactive behavior. This behavior was recovered in $Scn1a^{Stop/+}$-Cre animals (Fig. 4b, c). We also analyzed the length of time spent in the three areas in which the arena was divided (home, transition, exploration) (Fig. 4a), as anxious mice are expected to spend more time in the home and less time in the center. No difference among $Scn1a^{+/+}$ and $Scn1a^{Stop/+}$ was reported, suggesting normal anxiety levels in this DS mouse model (Fig. 4d). However, when an unfamiliar object was placed in the center of the arena, $Scn1a^{Stop/+}$ mice spent a significant higher amount of time in close proximity and sniffing the novel object compared to control mice, suggesting increased curiosity and/or impulsive

behavior (Fig. 4e–g). Importantly, mice in which $Scn1a$ gene was reactivated at P30 displayed a normalization of these features (Fig. 4e–g).

To examine social behavior, we performed the three-chamber test, in an arena with a central area connected with two lateral chambers each of which containing a mouse-sized wire cage (Fig. 4h). We first performed a 10 min habituation trial, with experimental mice freely moving in all the three areas of the arena and empty cages located in the right and the left chambers (Supplementary Fig. 5e). While $Scn1a^{+/+}$ and $Scn1a^{Stop/+}$-Cre mice showed no preference for the two zones with empty cages, $Scn1a^{Stop/+}$ displayed a preference for zone 1 (Supplementary Fig. 5f). Then, when a stranger age- and sex-matched mouse was added in one of the cages, $Scn1a^{+/+}$ mice spent more time in the mouse-containing chamber than in the empty cage-containing chamber and interacted extensively with peer mice (Fig. 4i). In contrast, $Scn1a^{Stop/+}$ mice showed no preference for the chamber hosting the stranger mouse (Fig. 4i), and their sociability index (SI), calculated as the ratio between the time spent sniffing the unfamiliar mouse and the total time spent sniffing the mouse and the empty cage is decreased in comparison to control (Fig. 4j, k). These observations suggested that $Scn1a^{Stop/+}$ mice have a lower level of sociability. In addition, their hyperactive behavior was confirmed (Supplementary Fig. 5g, h). On the contrary, $Scn1a^{Stop/+}$-Cre mice showed higher preference for the chamber with a novel unfamiliar mouse to the empty one (Fig. 4i), and they show a trend toward the recovery of the SI (Fig. 4j, k), suggesting that social interaction deficits can be rescued once $Scn1a$ allele expression is restored.

To test cognitive functions, the 8-arm radial and water mazes were employed to assess working memory and spatial reference learning memory, respectively. In the radial maze test, mildly food-deprived mice were allowed to explore the maze and consume all eight food rewards for a maximum time of 10 min (Fig. 5a). Mice were considered to make a mistake when either they enter a previously visited arm or when they do not consume the food reward. Remarkably, in the first two days of test, $Scn1a^{Stop/+}$ showed a higher number of entries in radial maze arms compared to $Scn1a^{+/+}$, correlating with previously observed hyperactivity in a novel environment (Fig. 5b). However, over the 10 day trial, control $Scn1a^{+/+}$ mice progressively learned to patrol the maze while, conversely, significantly more mistakes were made by $Scn1a^{Stop/+}$ mice (Fig. 5c).

In addition, the number of consecutive correct arm choices remained under the chance level (set at 5.5) during all the trials, indicating a defect in working memory in $Scn1a^{Stop/+}$ mice (Fig. 5d). On the contrary, $Scn1a^{Stop/+}$-Cre mice could overcome the chance level already at the 5th day of test indicating a complete recover of this defect (Fig. 5d). In the water maze test, during the acquisition phase, the mice had to learn the position of a hidden platform using spatial cues, while in the reversal phase, the position of the platform was changed, and animals needed to learn the new location (Fig. 5e). $Scn1a^{Stop/+}$ mice showed an increased escape latency compared to $Scn1a^{+/+}$ in both acquisition and reversal phases, which was normalized in $Scn1a^{Stop/+}$-Cre (Fig. 5f). Furthermore, we looked at the different strategies that mice adopt to reach the platform (Fig. 5g, h). In the acquisition phase, the wall hugging (WH) strategy was more exploited by $Scn1a^{Stop/+}$ mice, suggesting a problem-solving deficit[18] with respect to $Scn1a^{+/+}$ and $Scn1a^{Stop/+}$-Cre (Fig. 5g) that instead adopted more frequently the direct swim (SP) in both the acquisition and reversal phase (Fig. 5h).

Altogether, these results reveal that normalization of $Na_v1.1$ protein level after symptom onset is sufficient to rescue the main behavioral alterations characteristic of DS mice including

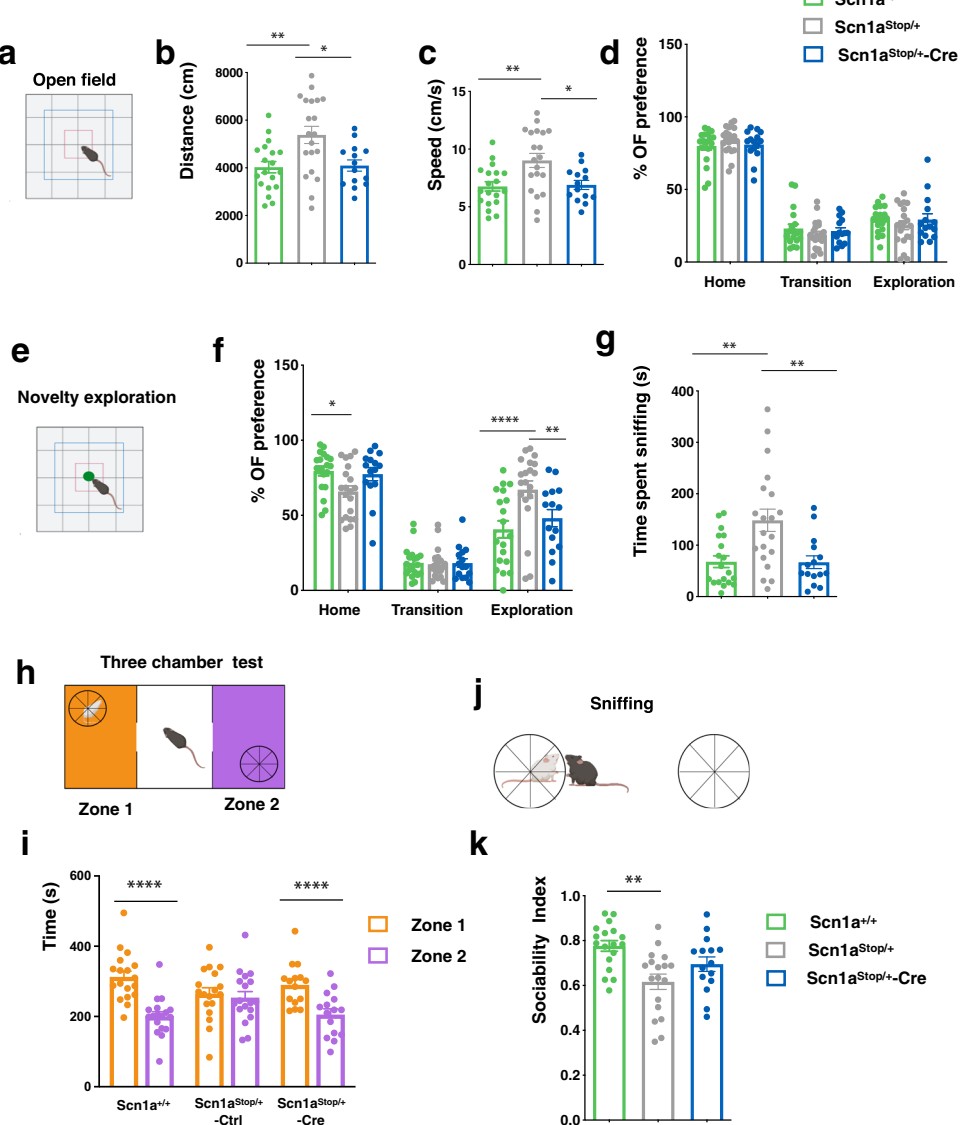

**Fig. 4 Re-expression of *Scn1a* gene after symptom onset rescues altered interactions with environment and social impairment in Dravet mice.**
**a** Scheme of the open field arena (**b**) distance traveled (**p = 0.0041, *p = 0.0134, one way-ANOVA followed by Tukey's multiple comparisons). **c** Velocity
(**p = 0.0048, *p = 0.016, one way-ANOVA followed by Tukey's multiple comparisons). **d** Percentage of time spent in home (two way-ANOVA).
**e** Scheme of the arena with a novel object in the center. **f** Percentage of time spent in home, transition and exploration areas after that a novel object has
been placed in the center of the arena; (*p < 0.05, **p = 0.006, ****p < 0.0001, two way-ANOVA followed by Tukey's multiple comparisons). **g** Time spent
sniffing the novel object (**p < 0.005, one way-ANOVA with Tukey's). n = 19 for *Scn1a*$^{+/+}$, n = 20 for *Scn1a*$^{Stop/+}$-Ctrl and n = 15 for *Scn1a*$^{Stop/+}$ -Cre for
open field test. **h** Scheme of the three chamber test. **i** Time spent in zone1 and zone 2 chambers when an unfamiliar mouse is placed in zone 1 chamber
(****p < 0.0001, two-way ANOVA followed by Tukey's multiple comparisons). **j** Sniffing. **k** Sociability index. (**p = 0.0010, one-way ANOVA followed by
Tukey's multiple comparisons). Data are shown as mean ± SEM, with dots representing individual mice, n = 18 for *Scn1a*$^{+/+}$, n = 18 for *Scn1a*$^{Stop/+}$-Ctrl and
n = 15 for *Scn1a*$^{Stop/+}$ -Cre for three-chamber test. Source data are provided as a Source Data file. **a**, **e**, **h**, **j** were Created with BioRender.com.

hyperactivity, social interaction, working memory and spatial
reference memory defects.

**Scn1a gene reconstitution rescued *Scn1a*$^{Stop/+}$ PV interneuron
firing failure**. Impairment in excitability and high-frequency
firing of different subtypes of GABAergic interneurons is a
characteristic of DS mice[13,14,19–21], with a more prominent defect
in Parvalbumin$^+$ (PV) fast spiking (FS) interneurons[13,14]. To
assess if this defect characterizes also our reversible mouse model
and drives the characteristic associated phenotype, we performed
whole-cell patch clamp analysis on acute brain slices of *Scn1a*$^{Stop/+}$ (129 Sv) mice crossed with Gad67-GFP knock-in mice[22]

(C57Bl6/N) first at P18-P23, when DS symptom start to develop.
Based on their firing pattern, GAD67-GFP + interneurons in
patched CA1 (Fig. 6b) were classified as FS or non-fast spiking
(non-FS) and analyzed separately.

Both FS and non-FS interneurons recorded in *Scn1a*$^{Stop/+}$ mice
show a reduced number of evoked action potentials (AP) at low and
high stimulation intensities, with a shift of input/output curves
toward higher current step amplitudes (Supplementary Fig. 7b, f).
Also maximal firing frequency was reduced in *Scn1a*$^{Stop/+}$ FS
interneurons with respect to control (Supplementary Fig. 7c), while
only a trend is observed in non-FS (Supplementary Fig. 7g).
Current threshold, defined as the minimal current injection able to
elicit neuronal firing, was increased in both interneuron subtypes

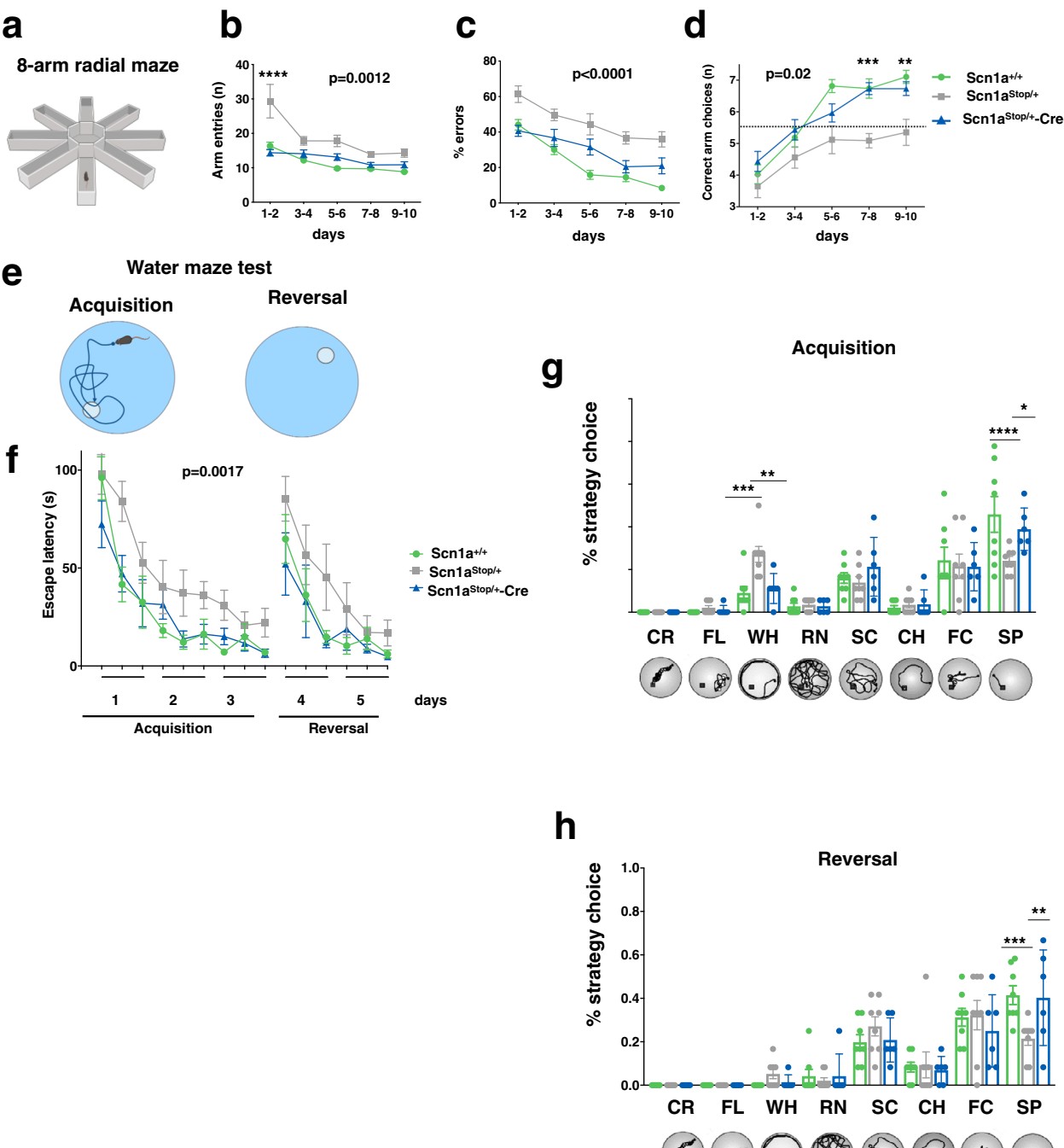

(Supplementary Fig. 7d, h). Other active and passive cell properties were not altered in $Scn1a^{Stop/+}$ interneurons (Supplementary Fig. S8a–n).

Interneuron firing defect was still detectable, although at a lesser extent, in $Scn1a^{Stop/+}$ mice at P28-P33, that is the time when we reactivate $Na_v1.1$ expression. In fact, in both FS and non-FS interneurons the difference between $Scn1a^{+/+}$ and $Scn1a^{Stop/+}$ appeared more evident only at current steps of higher amplitude (Supplementary Fig. S7j, k, n, o) and no difference in current threshold was detected (Supplementary Fig. S7l, p). While no difference in active and passive properties of non-FS interneurons was observed between the two experimental conditions (Supplementary Fig. 8p, r, t, v, x, z, b'), FS interneurons showed increased input resistance (Supplementary Fig. 8s) reduced maximal AP rise and decay slope (Supplementary Fig. 8y, a').

To understand if the rescue in both seizure and behavioral comorbidities, that we reported upon reactivation of $Na_v1.1$, was supported by a recovery in the firing ability of interneurons, we also performed whole-cell patch-clamp analysis on acute brain slices at P45-P55 of $Scn1a^{Stop/+}$;Gad67-GFP knock-in mice infused at P30 with the control and Cre viruses (Fig. 6a). Differently from PV interneurons of the neocortex of DS mice[10], FS interneurons in CA1 of $Scn1a^{Stop/+}$ still showed a decreased firing rate at high stimulation intensity compared to $Scn1a^{+/+}$ (Fig. 6c, d, e), indicating that they retained the excitability impairment also at a late stage of the disease. Interestingly, at this developmental stage, $Scn1a^{Stop/+}$-Cre FS interneurons recovered this firing defect (Fig. 6d) and appeared even more excitable than control FS interneurons at lower stimulation intensity (Fig. 6c, d, e). A reduction in current threshold, corroborated

**Fig. 5 Restoration of Na$_v$1.1 expression after symptom onset rescues cognitive impairment in DS mice. a** Radial maze test. Scheme of the eight-arm radial maze test. **b** Plot of total number of entries in radial maze arms; 1–2 days: $Scn1a^{Stop/+}$ vs. $Scn1a^{+/+}$ and vs. $Scn1a^{Stop/+}$;Cre ****$p < 0.0001$; 3–4 days: $Scn1a^{Stop/+}$ vs. $Scn1a^{+/+}$ *$p = 0.025$; 5–6 days: $Scn1a^{Stop/+}$ vs. $Scn1a^{+/+}$, ***$p = 0.0006$; 9–10 days: $Scn1a^{Stop/+}$ vs. $Scn1a^{+/+}$ *$p = 0.0334$. **c** Plot of percentage of errors (entries in arms already visited) on total visits; (1–2 days: $Scn1a^{Stop/+}$ vs. $Scn1a^{+/+}$ **$p = 0.024$ and vs. $Scn1a^{Stop/+}$-Cre ***$p = 0.0005$; 3–4 days: $Scn1a^{Stop/+}$ vs. $Scn1a^{+/+}$ ***$p = 0.0004$ and vs. $Scn1a^{Stop/+}$-Cre *$p = 0.049$; 5–6 days: $Scn1a^{Stop/+}$ vs. $Scn1a^{+/+}$, ****$p < 0.0001$ and $Scn1a^{+/+}$ vs. $Scn1a^{floxSTOP/+}$;**$p = 0.0092$; 7–8 days: $Scn1a^{Stop/+}$ vs. $Scn1a^{+/+}$ ****$p < 0.0001$ and vs. $Scn1a^{Stop/+}$-Cre **$p = 0.0083$; 9–10 days: $Scn1a^{Stop/+}$ vs. $Scn1a^{+/+}$ ****$p < 0.0001$ and vs. $Scn1a^{Stop/+}$-Cre *$p = 0.0177$. **d** Plot of the number of consecutive correct arm choices; 5–6 days: $Scn1a^{Stop/+}$vs $Scn1a^{+/+}$ *** $p = 0.0002$; 7–8 days: $Scn1a^{Stop/+}$vs. $Scn1a^{+/+}$ *** $p = 0.0003$, $Scn1a^{Stop/+}$ vs. $Scn1a^{Stop/+}$-Cre ***$p = 0.0008$; 9–10 days: $Scn1a^{Stop/+}$ vs. $Scn1a^{+/+}$ ***$p = 0.0001$, $Scn1a^{Stop/+}$ vs. $Scn1a^{Stop/+}$-Cre **$p = 0.0065$. **e** Two-way ANOVA with Sidak's multiple comparisons $Scn1a^{+/+}$ $n = 19$ $Scn1a^{Stop/+}$ $n = 17$; $Scn1a^{Stop/+}$-Cre $n = 15$). **f** Water maze test. Scheme of the acquisition and reversal phase of the test. Escape latency during acquisition and reversal phases of the test, two way ANOVA $p = 0.0017$ followed by Bonferroni/ Dunn multiple comparison $Scn1a^{Stop/+}$vs. $Scn1a^{+/+}$ $p = 0.016$; $Scn1a^{Stop/+}$-Cre vs. $Scn1a^{Stop/+}$ $p = 0.0018$; $Scn1a^{Stop/+}$-Cre vs. $Scn1a^{+/+}$ $p = 0.8449$ **g** Percentage of mice classified according to their predominant search strategy of the platform during the acquisition, two way ANOVA followed by Turkey's multiple comparison WH: $Scn1a^{Stop/+}$ vs. $Scn1a^{+/+}$ ***$p = 0.0005$; $Scn1a^{Stop/+}$-Cre vs $Scn1a^{Stop/+}$ **$p = 0.005$; SP: $Scn1a^{Stop/+}$ vs. $Scn1a^{+/+}$ ****$p < 0.0001$; $Scn1a^{Stop/+}$-Cre vs. $Scn1a^{Stop/+}$ *$p = 0.014$. **h** Reversal phase SP: $Scn1a^{Stop/+}$ vs. $Scn1a^{+/+}$ ***$p = 0.003$; $Scn1a^{Stop/+}$-Cre vs. $Scn1a^{Stop/+}$ **$p = 0.0018$, $Scn1a^{+/+}$ $n = 8$, $Scn1a^{Stop/+}$ $n = 8$, $Scn1a^{Stop/+}$-Cre $n = 7$. CR Circling, FL Floating, WH Wall hugging, RN Random swimming, SC Scanning, CH Chaining, FC Focal searching, SP Direct swims. Source data are provided as a Source Data file. **a–e** were Created with BioRender.com.

these observations (Fig. 6f). No difference was observed in passive and other active cell properties between the three experimental groups (Supplementary Fig. 9).

We observed a different scenario for non-FS Gad67-GFP$^+$ interneurons at P45-P55. In fact, $Scn1a^{Stop/+}$ interneuron excitability was comparable to those of control $Scn1a^{+/+}$ (Fig. 6g–j), while $Scn1a^{Stop/+}$-Cre interneurons resulted more excitable with respect to both $Scn1a^{+/+}$ and $Scn1a^{Stop/+}$, in particular at lower stimulation intensity (130-440 pA current steps) (Fig. 6g, h). Maximal AP frequency was comparable in $Scn1a^{+/+}$ and $Scn1a^{Stop/+}$ Gad67-GFP$^+$ non-fast spiking CA1 interneurons, while a trend toward higher frequency was reported in interneurons from $Scn1a^{Stop/+}$-Cre compared to the other groups (Fig. 6i). Current thresholds showed an increased excitability in neurons from $Scn1a^{Stop/+}$-Cre mice compared to those from $Scn1a^{+/+}$ and $Scn1a^{Stop/+}$ littermates (Fig. 6j).

Analysis of passive cell properties showed a reduction in cell capacitance in $Scn1a^{Stop/+}$-Cre Gad67-GFP$^+$ non-FS CA1 interneurons compared to $Scn1a^{Stop/+}$-Ctrl. Input resistance appeared also to be increased in $Scn1a^{Stop/+}$-Cre interneurons compared to $Scn1a^{Stop/+}$ (Supplementary Fig. 9b, f). We did not report any changes in the other passive and active cells properties analyzed (Supplementary Fig. 9).

To assess if increased excitability of CA1 interneurons due to Na$_v$1.1 normalization produced an increase of the inhibitory input on pyramidal neurons, we recorded IPSCs from CA1 pyramidal neurons. First, a decrease of IPSC frequency onto pyramidal neurons in $Scn1a^{Stop/+}$ mice compared to $Scn1a^{+/+}$ was observed (Fig. 6k, k'l), with no significant difference in the amplitude. Cre-mediated reactivation of $Scn1a$ recovered the frequency of IPSCs (Fig. 6k, l, m), but altered their amplitude distribution with a selective increase of small amplitude IPSCs (<20 pA) and a decrease of high amplitude events (>20 pA) (Supplementary Fig. 9p, q).

Altogether, we observed a global increase in the excitability of CA1 interneurons in DS mice upon $Scn1a$ gene reactivation that can induce an increase of small amplitude IPSCs onto pyramidal neurons.

**Profound alterations in gene expression of Scn1a$^{Stop/+}$ mice are rescued upon Na$_v$1.1 level restoration**. Alterations in global gene expression are associated to epilepsy[23–27] and have been reported in genetic models of epilepsy, including DS mouse models[28–30]. To determine the transcriptional alterations in $Scn1a^{Stop/+}$ mice and to which extent were rescued after Scn1a

gene restoration, we processed the three experimental groups of mice at the end of behavioral studies (4–5 month old) for transcriptomics analysis by bulk RNA-seq using isolated cortical and hippocampal tissues (Supplementary Fig. 10a). t-distributed stochastic neighbor embedding (t-SNE) analysis evidenced a good clustering between cortical and hippocampal tissues (Supplementary Fig. 10b) and among relative tissue genotypes (Supplementary Fig. 10c, d).

In $Scn1a^{Stop/+}$-Ctrl respect to $Scn1a^{+/+}$ cortical tissues a total of 1198 differentially expressed genes (DEGs) (809 upregulated and 389 downregulated) were identified while, surprisingly, no DEG was found in $Scn1a^{Stop/+}$-Cre versus $Scn1a^{+/+}$ genotypes. Additionally, 1417 DEGs were detected between $Scn1a^{Stop/+}$-Ctrl and $Scn1a^{+/+}$ hippocampal tissues (786 upregulated and 631 downregulated) (Supplementary Data 1 and Supplementary Fig. 10e) and the majority of them were rescued in $Scn1a^{Stop/+}$-Cre hippocampi ($Scn1a^{Stop/+}$-Cre versus $Scn1a^{Stop/+}$-Ctrl total DEGs: 163 with 111 genes upregulated and 52 downregulated) (Supplementary Fig. 10f). Interestingly, GO categories of DEGs in both cerebral cortex and hippocampus highlighted alterations of functions and pathways that are relevant to DS phenotype including sodium ion transport, regulation of synapses, cytoskeleton reorganization, extracellular matrix disassembly, memory and learning, effects on angiogenesis but also response to hypoxia, ischemia and inflammatory response (Fig. 7a and Supplementary Data 2).

Inflammation is a well-known pathological correlate of epileptic activity[31–33] and it has been recently described also in DS mice[34]. RNA-seq dataset analysis showed up-regulation of generic neuroinflammation and pan-reactive astrocytic markers, including glial fibrillary acidic protein (GFAP), vimentin (Vim) in both $Scn1a^{Stop/+}$-Ctrl cortical and hippocampal tissues, with more profound alterations in the latter samples (Fig. 7a, b, c). Two different types of reactive astrocytes have been described: "A1", that activate complement cascade genes and have a neurotoxic effect; "A2", that instead secrete neurotrophic factors and play a protective role on neurons[35,36]. Although not all the markers of A2 astrocytes are significantly up regulated in $Scn1a^{Stop/+}$ mice, the proportion of genes up regulated in A2 category is higher than those in A1 group (Fig. 7d, e and Supplementary Data 1), suggesting that DS astrocytes have a more evident A2 phenotype. Quantification of GFAP$^+$/GS$^+$ (glutamine synthetase) and Vimentin+ astroglial cells in the dentate gyrus (DG) of the hippocampus of 8 week-old mice confirmed increased reactive astrocytosis in $Scn1a^{Stop/+}$-Ctrl respect to $Scn1a^{+/+}$ mice (Fig. 7f–g", i–j', l–n), in agreement with

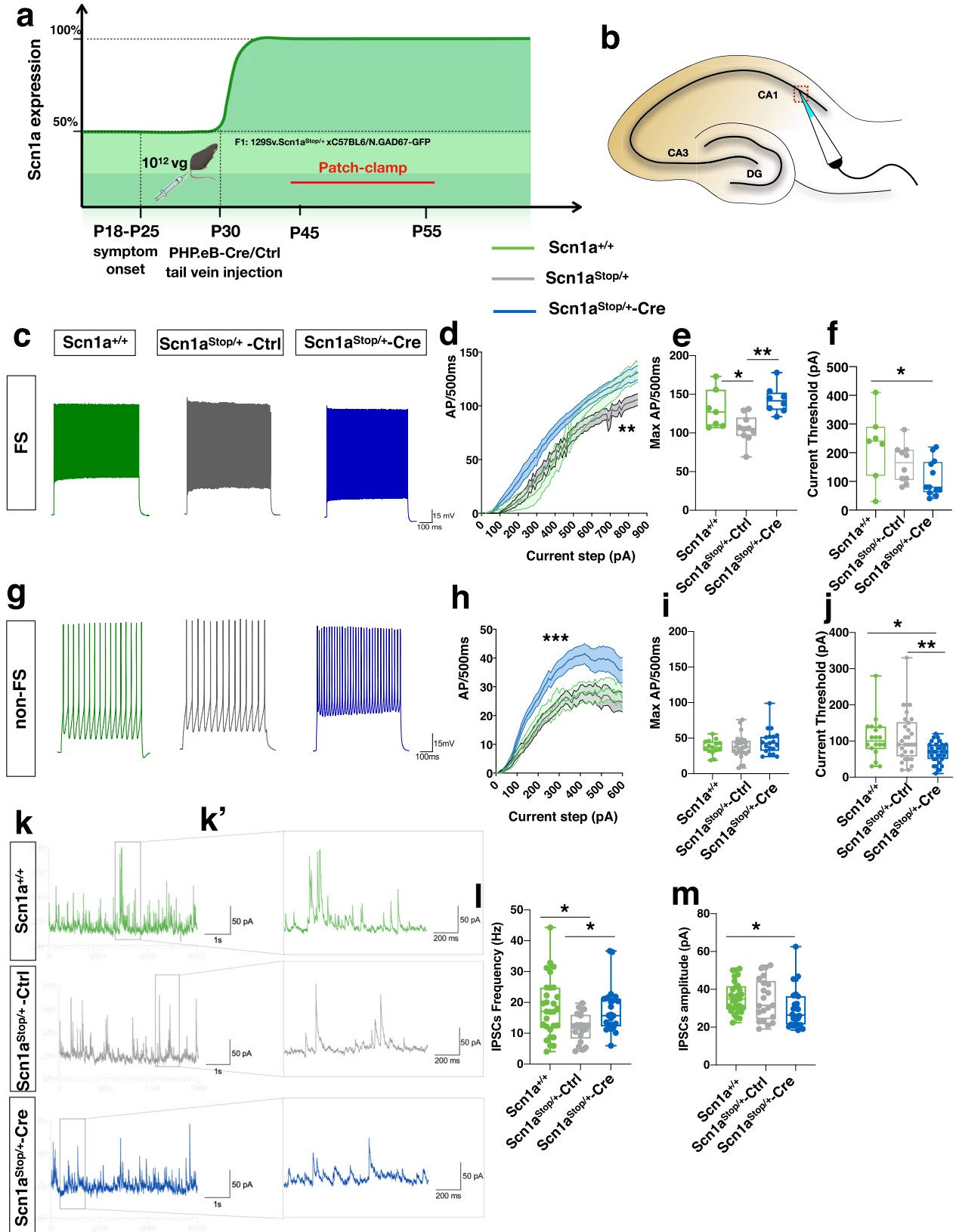

previous observations[34]. In addition, GFAP+/GS+ cells showed increased morphological complexity, measured by 3D-Sholl analysis, in DS animals versus control (Fig. 7m). Inflammation-related genes were rescued in DS mice after Na$_v$1.1 level normalization (Fig. 7b–e) and immunohistochemical analysis in the DG of Scn1a$^{Stop/+}$-Cre mice confirmed those data (Fig. 7h–n).

No significant alterations in inflamed microglia-specific genes were detected (Supplementary Fig. 12a). In fact, quantification of the total Iba1+ microglial cells and Cd68 + /Iba1 + cells in the

**Fig. 6 FS and non-FS interneuron activity after Na$_v$1.1 re-expression. a** Timeline for patch-clamp recording experiments. $Scn1a^{Stop/+}$ mice were injected with either PHP.eB-Ctrl or PHP.eB-Cre at P30 and patched between P45 and P55. **b** Scheme of CA1 where GAD67-GFP + neurons were patched. **c** Representative traces of fast-spiking interneurons in CA1 of $Scn1a^{+/+}$, $Scn1a^{Stop/+}$ -Ctrl and $Scn1a^{Stop/+}$-Cre mice in response to 800pA 500 ms current step. **d** Average firing rates in response to different current steps for FS GAD67-GFP + neurons in CA1 of the three experimental groups. ($n = 7$ for $Scn1a^{+/+}$ 4 animals, $n = 10$ for $Scn1a^{Stop/+}$ -Ctrl 6 animals and $n = 12$ for $Scn1a^{Stop/+}$-Cre 5 animals, **$p = 0.0060$, Two-way ANOVA;). **e** Maximal firing rate extrapolated from I/O curve, (*$p = 0.0474$, **$p = 0.0021$, one-way ANOVA, with Tukey's post hoc comparison). **f** Current threshold, (*$p = 0.0246$, One-way ANOVA, with Tukey's post hoc comparison). **g** Representative traces from non -FS Gad67-GFP$^+$ interneurons in CA1 of $Scn1a^{+/+}$, $Scn1a^{Stop/+}$ -Ctrl and $Scn1a^{Stop/+}$-Cre in response to 200 pA current step. **h** Average firing rates in response to different current steps for non-FS GAD67-GFP + interneurons in CA1 of the three experimental groups ($n = 18$ for $Scn1a^{+/+}$ 4 animals, $n = 30$ 6 animals for $Scn1a^{Stop/+}$ -Ctrl and $n = 32$ for $Scn1a^{Stop/+}$-Cre 5 animals, (***$p = 0.0008$, two-way ANOVA repeated measures with Tukey's post hoc comparison). **i** Maximal AP frequency of non -FS GAD67-GFP + interneurons (one-way ANOVA with Tukey's post hoc comparison). **j** Current threshold of non-FS GAD67-GFP + interneurons. *$p = 0.0336$, **$p = 0.0066$, one-way ANOVA, with Bonferroni post hoc comparison). **k–k'** Representatives IPSCs recorded from CA1 pyramidal neurons and relative magnifications. **l** IPSC frequency ($n = 28$ for $Scn1a^{+/+}$ 6 mice, $n = 21$ for $Scn1a^{Stop/+}$ -Ctrl 4 mice and $n = 26$ for $Scn1a^{Stop/+}$-Cre 4 mice) (*$p = 0.0115$, One-way ANOVA, with Tukey's post hoc comparison). **m** IPSC amplitude (n = 28 for $Scn1a^{+/+}$ 6 mice, $n = 21$ for $Scn1a^{Stop/+}$ -Ctrl 4 mice and $n = 26$ for $Scn1a^{Stop/+}$-Cre 4 mice) (*$p = 0.0483$, One-way ANOVA, with Tukey's post hoc comparison). Data shown are means ± SEM for **d** and **h**; for box plot in **e**, **f**, **i**, **j**, **l** and **m** each dot represent mean values from each cell, central lines median value and box limits represent 25% and 75% percentiles, while whiskers minimal and max values. Source data are provided as a Source Data file.

DG of the hippocampus of 8 week-old mice did not highlight any significant change between control and DS mouse brain, confirming that microglia is not characterized by an inflammatory state in our DS model (Supplementary Fig. 12b, c, d).

Overall, these results revealed the presence of reactive astrocytes with a more pronounced A2 gene expression profile in $Scn1a^{Stop/+}$ mice indicating the polarization of astrocytes toward a relatively protective state rather than to an A1 neurotoxic phenotype. Importantly, the majority of gene expression alterations were recovered in $Scn1a^{Stop/+}$-Cre mice.

**Normalization of Na$_v$1.1 levels in adult $Scn1a^{Stop/+}$ mice (P90) rescues the seizure phenotype.** Once assessed complete symptomatic reversibility upon restoration of Na$_v$1.1 levels in juvenile DS mice, we sought to determine if it can be achieved also in adult $Scn1a^{Stop/+}$ mice after a prolonged pathological period. To this aim, we triggered Na$_v$1.1 re-activation in 3-month-old mice (P90), roughly 10 weeks after initial disease manifestations (Fig. 8a). $Scn1a^{Stop/+}$ mice were implanted with EEG transmitters at around P70 and, few days after were video-EEG recorded for 2 weeks (baseline). Mice that experienced at least one seizure over the 2 weeks of recordings were randomized and were administrated with either PHP.eB-Cre or PHP.eB-Ctrl through tail vein injections at P90. Thereafter, video-EEG continued for further 25 days, until mice reached postnatal day 115 (Fig. 8a). Then, to investigate the ability of Na$_v$1.1 expression restoration to treat chronic epilepsy in adult $Scn1a^{Stop/+}$ mice, we analyzed the frequency of generalized tonic-clonic seizures in each animal before and after viral administration (Fig. 8b). We first confirmed that there was no significant difference in seizure frequencies during the baseline period between the two groups. $Scn1a^{Stop/+}$-Ctrl mice show no difference in the seizure frequency after the viral injection in comparison to baseline, with 80% of mice showing more seizures with respect to baseline (Fig. 8b, c). Conversely, $Scn1a^{Stop/+}$-Cre mice (6 out of 6) showed a significant reduction in the number of seizures in the first 8 days after viral treatment and completely disappeared in the following days with 100% of mice showing seizure reduction in comparison to baseline (Fig. 8b, c). At the end of video-EEG sessions, mice were subjected to hyperthermia induced seizures. All $Scn1a^{Stop/+}$-Ctrl mice developed a tonic clonic seizure when body temperature was raised up to 42.5 °C, conversely only one $Scn1a^{Stop/+}$-Cre animal developed a tonic clonic seizure at 42.4 °C (Fig. 8e). Viral copy number analysis evidenced a prevalent targeting of the PHP.eB-Cre in the cerebral cortex respect to the liver (Supplementary Fig. 13a) and Western blot analysis evidenced that viral

biodistribution in the brain of adult mice was sufficient to regain normal levels of Na$_v$1.1 protein in cerebral cortex and hippocampus but not in cerebellum and striatum (Supplementary Fig. 13b, c). Nevertheless, Na$_v$1.1 restoration in cerebral cortex and hippocampus was sufficient to rescue the epileptic phenotype.

Taken together, these results showed that normalization of Na$_v$1.1 level is sufficient to recover the DS epileptic phenotype also in adult mice, ten weeks after the disease onset.

## Discussion
Herein we generated and characterized a DS mouse model where the $Scn1a$ silent allele was conditionally reactivated at different time points during disease progression. While numerous therapeutic approaches to enhance $Scn1a$ gene expression levels are in development or at early stages of clinical trial for treating DS patients, whether and to which extent normalization of Na$_v$1.1 level can rescue the disease phenotype is currently unknown. As a consequence, if those treatments do not show the desired effect on the pathology, it will remain unclear whether only technical limitations or the real impossibility to recover DS manifestations are responsible for the failure or reduced efficacy of those strategies. Hence, investigating DS reversibility in a condition where Na$_v$1.1 protein level can be properly and finely controlled is of crucial relevance to fill this gap of knowledge.

In this study, we showed that efficient restoration of $Scn1a$ gene expression in a DS mouse model after symptom onset, can suppress seizures together with the associated severe behavioral alterations including hyperactivity, social interaction impairment and cognitive defects.

Since Dravet mouse models start to develop symptoms around P18-P24, we proceeded with gene reactivation at P30. In this way, we selected a cohort of mice that survived the critical time window in which most of the SUDEP occur and that may present a milder phenotype. However, this time frame in mice corresponds roughly to 4 years in humans[37,38], when, after consolidated diagnosis of DS, patients would be eventually ready for gene therapy administration.

In the $Scn1a^{Stop/+}$ model, $Scn1a$ gene re-activation is dependent on excision of the STOP cassette which can be achieved by genetic crossing with Cre transgenic mice. We confirmed that after crossing with CMV-Cre mice, the silent $Scn1a$ allele was fully re-expressed reaching physiological Na$_v$1.1 protein level. However, in the subsequent analysis the Cre expression was vehiculated by viral gene transfer. AAV-PHP.eB synthetic capsid offers the unprecedented opportunity to transduce the whole

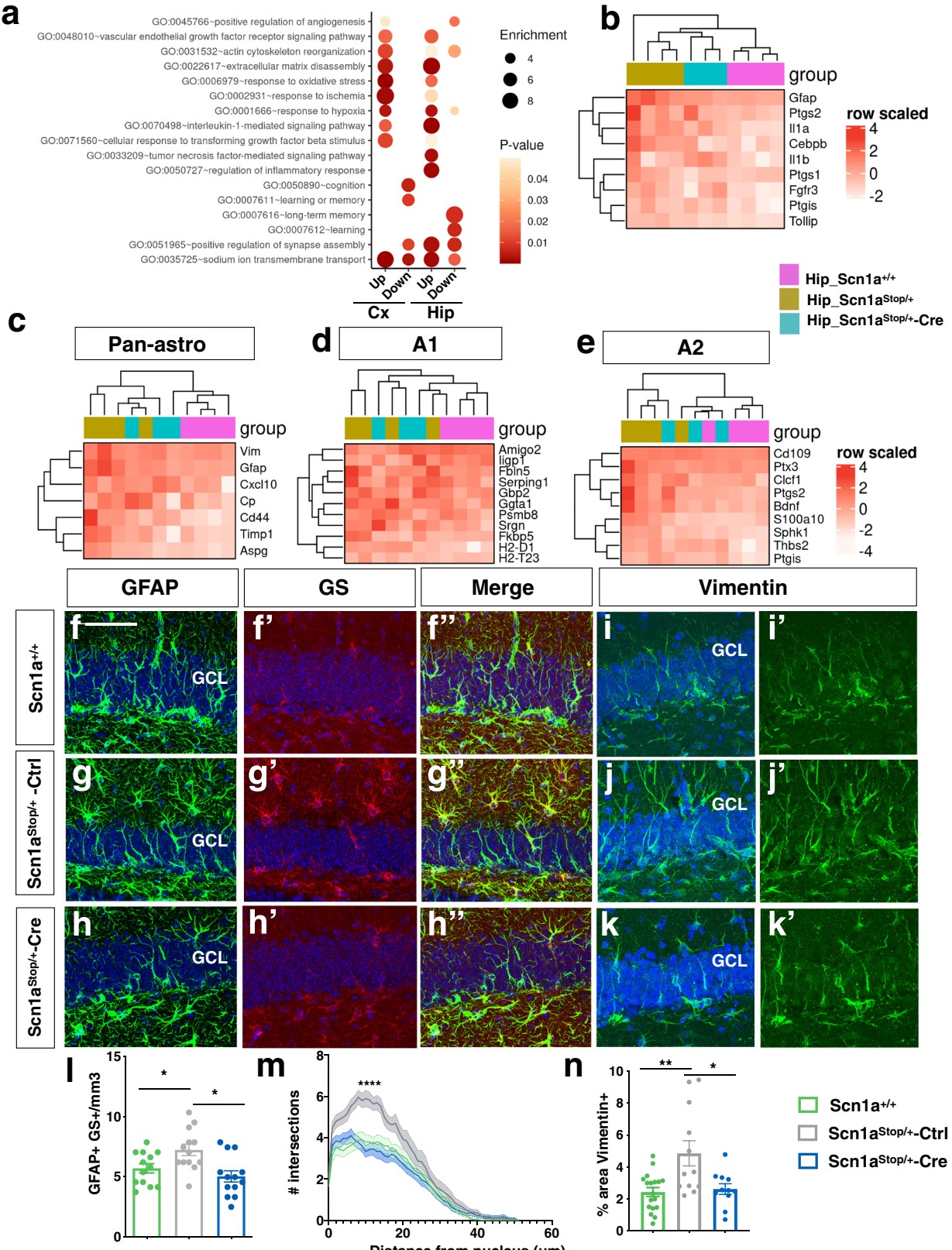

brain by peripheral intra-vascular administration[15,39], an ideal route also for treating DS patients. This approach, although not based on the delivery of a proper gene therapy, offered the possibility to mimic that treatment and its related challenges such as virus titration, time required for viral gene expression, and biodistribution in different brain areas.

The administration of the Cre-expressing virus was sufficient to rescue both spontaneous and febrile seizures. Moreover, we observed that suboptimal *Scn1a* gene re-expression was sufficient to protect from spontaneous seizures, but not to prevent SUDEP following thermo-induction. This aspect warrants further investigation since understanding of the minimal number of neurons

**Fig. 7 DS mouse brain astrocytosis is recovered upon Na$_v$1.1 restoration. a** Main canonical pathways identified by IPA that are differentially expressed between $Scn1a^{+/+}$ and $Scn1a^{Stop/+}$ mice. **b** Heatmaps showing the different level of expression of selected genes related to neuroinflammation in $Scn1a^{+/+}$, $Scn1a^{Stop/+}$ -Ctrl and $Scn1a^{Stop/+}$-Cre mouse hippocampi. **c** Heatmaps showing the different level of expression of makers of pan-astrocytosis. **d** A1 type reactive astrocytes, **e** A2 type reactive astrocytes in the 3 experimental groups. **f–h** Representative images of immunofluorescence for GFAP and GS in DG $Scn1a^{+/+}$, $Scn1a^{Stop/+}$ and $Scn1a^{Stop/+}$ -Cre mouse hippocampi. **i–k** Representative images of immunofluorescence for Vimentin (Vim) in DG of $Scn1a^{+/+}$, $Scn1a^{Stop/+}$ -Ctrl and $Scn1a^{Stop/+}$ -Cre mouse hippocampi. **l** Quantification of GFAP + /GS + cells (*$p < 0.05$, One-way ANOVA followed by Bonferroni's post-test, 13 brain sections for each group, 2 mice for genotype). **m** Sholl analysis of GFAP + /GS + cells (****$p < 0.0001$, two-way ANOVA, $n$ of cells = 15 for each group, 2 mice for genotype). **n** Percentage of CA1 area positive for Vimentin staining, (**$p = 0.0019$, *$p = 0.01$, one-way ANOVA followed by Tukey's post-test, brain sections $n = 18$ for $Scn1a^{+/+}$, $n = 12$ for $Scn1a^{Stop/+}$ and $n = 11$ for $Scn1a^{Stop/+}$ -Cre, 2 mice for each genotype). Scale bars, 50 μm. Data are means ± SEM. Source data are provided as a Source Data file.

to be corrected to achieve complete symptomatic recovery is crucial for guiding the ongoing efforts to design therapeutic approaches.

DS is associated with a range of behavioral disorders, together with motor and cognitive issues that persist also in those cases in which a discrete control of seizures is achieved, and strongly affect life of the patients. The first years of development in DS patients are considered normal[40], while comorbidities have always been associated to the onset of severe epilepsy during the worsening stage, suggesting that seizure severity has a strong impact on disease manifestations[1]. More recently, other reports suggested that developmental delay may be independent of epilepsy[41,42] and subtle developmental changes have been described during the febrile stage[43,44]. In line with this, a recent study showed that in DS mice motor impairment and hyper-activity are evident already in the febrile stage (P14-P18) while deficits in working memory emerged during the worsening stage (P21-P27)[45]. While in our model we did not detect any motor alteration at the age of analysis (3–4 months), hyperactivity, and cognitive deficits were evident and Na$_v$1.1 reactivation at P30 was sufficient to rescue them.

Such an extensive symptom reversion was not easily pre-dictable a priori. In fact, a recent study evidenced that PV$^+$ interneurons of layer 2/3 of somatosensory cortex recover the firing defect characteristic of critical period in a DS model at later developmental stages (P35-P55)[10], although mice con-tinue to seize and to show behavioral abnormalities. Those results might imply that PV$^+$ interneuron dysfunctions in the critical period provoke alterations in neuronal circuits that contribute to the phenotype in chronic phases of DS. Alter-natively, defects in other interneuron subclasses (SST$^+$, CCK + and VIP$^+$) and/or PV$^+$ interneurons resident in other brain areas may maintain their dysfunctional state up to adulthood[10]. To distinguish among those possibilities, we performed patch-clamp recordings on Gad67-GFP$^+$ inter-neurons in the CA1 of the hippocampus, playing a pivotal role in seizure initiation and propagation[46] at different postnatal stages. Reduced interneuron excitability described in other DS mouse models affects both FS and non-FS interneurons in $Scn1a^{Stop/+}$mice at P18-P23, when first symptoms became evident. At P28-P33, interneuron defects were still evident in $Scn1a^{Stop/+}$ mice with respect to control, but differences were reduced in comparison to those observed in the previous time-point.

The analysis of brain slices from P44-P54 mice showed that FS Gad67-GFP$^+$ interneurons in this region still presented a reduced firing frequency in $Scn1a^{Stop/+}$ mice, differently from what has been observed in the cerebral cortex[10] and confirming that FS interneurons from different brain areas might retain pathological alterations for different amount of time. Conversely, Gad67-GFP$^+$ non-FS interneurons in CA1 of DS mice, showed an unex-pected complete recovery in excitability. The progressive nor-malization of interneuron activity that we started to observe at

P28-P33 in $Scn1a^{Stop/+}$ mice and that becomes more evident at P44-P54 might be determined by compensatory upregulation of other channel alpha subunits, such as Na$_v$1.2, 1.3, and 1.6, that have been previously hypothesized[10,13]. However, our bulk RNA-seq analysis did not detect any transcriptional alteration of the Na$_v$ alpha subunits in $Scn1a^{Stop/+}$ mice, thus post-transcriptional mechanisms might be responsible for this effect. For instance, a rebalance of the different Na$_v$ proteins between the intracellular and membrane pools can likely contribute to the observed functional compensation.

This change seems to be not repairable after physiological re-expression of Na$_v$1.1 in P30 $Scn1a^{Stop/+}$-Cre mice; in fact, we found an increased excitability in interneurons compared to controls: in particular, a recovery of FS interneuron excitability was evident at high stimulation, while an overall increase of non-FS interneuron excitability was observed. Probably, early inter-neuron impairment in DS mice, triggers a homeostatic response that induces a normalization of non-FS interneuron activity, and for this reason after Na$_v$1.1 re-expression at P30, we observed an increase in excitability above this compensation for both FS (only at low-intensity stimuli) and non-FS Gad67-GFP$^+$ CA1 interneurons.

Although the interneuron activity is not fully normalized, we observed a complete phenotypic recovery in DS animals upon Na$_v$1.1 restoration. In fact, our data suggest that reactivation of Na$_v$1.1 does not elicit complete recovery of all impairments, but it rather establishes an alternative excitatory/inhibitory balance sufficient to gain full phenotype normalization with rescue of both seizures and behavioral alterations. However, considering the time required to re-express physiological levels of Na$_v$1.1 after PHP.eB-Cre delivery, our patch-clamp analysis might have still detected an acute and transitory condition after Na$_v$1.1 reacti-vation. Thus, we can't exclude that complete rescue of intrinsic neuronal properties might be achieved, but it requires much longer time after Na$_v$1.1 restoration.

We also showed that rescue of the seizures can be achieved also reactivating Na$_v$1.1 expression in adult mice at P90. Considering that 3 month old mice correspond roughly to humans of 20 years of age[37,38], the results of this experiment help to look with optimism also to the treatment of adult patients, although other hurdles would need to be fully addressed such as the therapy dosage and delivery.

Bulk RNA transcriptomics performed in reversible DS and relative control animals unveiled dramatic changes in gene expression in both cerebral cortex and hippocampus and evi-denced an inflammatory status with up-regulation of pro-inflammatory cytokines, that was more evident in the hippo-campus. Markers of reactive astrogliosis were upregulated, con-firming previous observations in other DS mouse models[30,34]. Recent reports support the concept that reactive astrocytes can play both beneficial and detrimental roles depending on the nature of the injury or disease[35,36]. A2 reactive astrocytes, induced by ischemia, express high levels of neurotrophic factors,

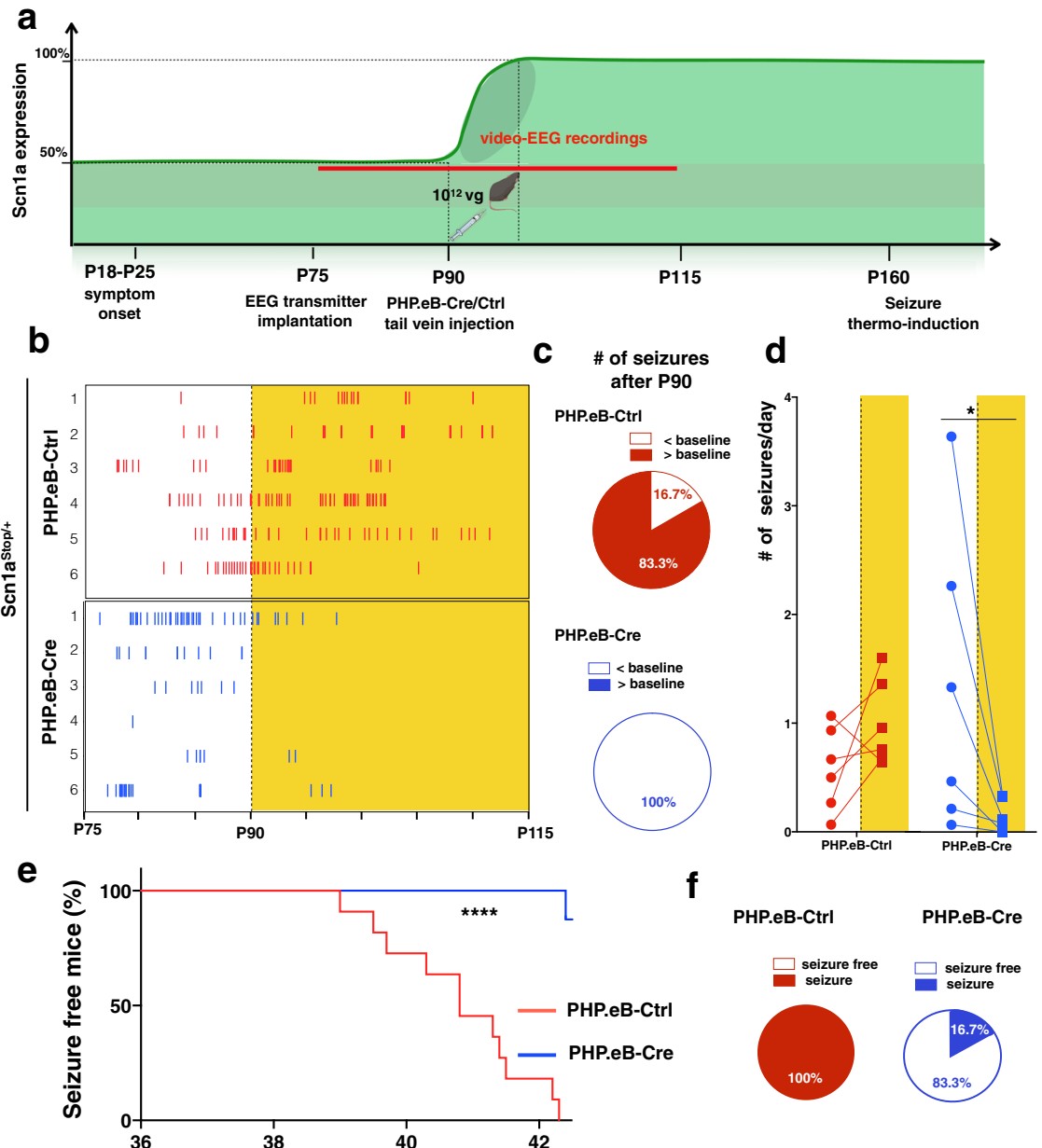

**Fig. 8 Re-expression of *Scn1a* gene in adult mice (P90) rescues seizures in DS mice. a** Schematics depicting the experimental timeline: *Scn1a*$^{Stop/+}$ mice were implanted with EEG transmitters at P70 and, few days after were video-EEG recorded for 2 weeks (baseline). Mice that experienced at least one seizure over the 2 weeks of recordings were randomized and were administrated with either PHP.eB-Cre or PHP.eB-Ctrl through tail vein injections at P90. Thereafter, video-EEG continued for further 25 days, until mice reached postnatal day 115. Created with BioRender.com. **b** Raster plot showing all generalized tonic-clonic seizures (Racine scale stages 4 and 5) in 6 *Scn1a*$^{Stop/+}$-Ctrl injected mice and 6 *Scn1a*$^{Stop/+}$-Cre mice **c** Pie chart showing the proportion of Ctrl- and -Cre-injected *Scn1a*$^{Stop/+}$ mice with an increased or decreased number of seizures compared to baseline after viral delivery ($p = 0.0152$; two-sided Fisher's exact test). **d** Seizure frequency during the baseline period and after viral injection in both Ctrl- and Cre-injected *Scn1a*$^{Stop/+}$ mice. Seizure frequency in the baseline between the two groups (*$p = 0.235$, t-test). Seizure frequency before and after viral administration in *Scn1a*$^{Stop/+}$-Ctrl ($p = 0.53$, two-way ANOVA with Sidak's multiple comparison, $n = 6$) and *Scn1a*$^{Stop/+}$-Cre; *$p = 0.028$, two-way ANOVA with Sidak's multiple comparison, $n = 6$), **e** Percentage of *Scn1a*$^{Stop/+}$-Ctrl and *Scn1a*$^{Stop/+}$-Cre mice remaining seizure-free after thermal induction ($n = 6$ for each group, ***$p = 0.0002$; Mantel-Cox log-rank test). **f** Pie charts showing the percentage of *Scn1a*$^{Stop/+}$-Ctrl and for *Scn1a*$^{Stop/+}$-Cre mice with or without induced thermal seizures ($p = 0.0152$; two-sided Fisher's exact test). Source data are provided as a Source Data file.

cytokines, and thrombospondins that may contribute to repair and rebuild lost synapses[47,48]. Conversely, A1 type reactive astrocytes, that are induced by inflamed microglia via secretion of Il-1α, TNF, and C1q[36,49], acquire a toxic phenotype contributing to synapse and neuron loss in neurodegenerative disease[36,50]. Intriguingly, reactive astrocytes in our DS mice presented a more pronounced A2 molecular signature, and to our knowledge, this

is the first time that a similar astrocytosis profile is associated with epilepsy. In agreement with this, no evident inflammatory state of microglia was detectable, and no neurodegeneration has been reported in DS patient brain tissue[51]. Thus, the A2 astrocytic polarization might have a pivotal role to support neuronal and synaptic survival and avoid neurodegeneration after an epileptic insult.

In conclusion, this study underlines the relevance of this $Scn1a^{Stop/+}$ DS mouse model to determine the extent of reversibility in DS at different times after symptom onset. These findings provide evidence that phenotype reversibility is possible in a mouse model of Dravet syndrome and may potentially be used to guide new therapeutic strategies.

## Methods

**Study design**. This study aimed to test the hypothesis that restoring physiological levels of Na$_v$1.1 in a DS mouse model is sufficient to revert the main features of the syndrome. For in vivo experiments the 3Rs guidelines for animal welfare were followed. Researchers were blinded during recordings and analysis.

**Mice**. Mice were maintained at San Raffaele Scientific Institute Institutional mouse facility (Milan, Italy) housed under 12 h dark-light cycle, with a relative humidity of 50–60%, a temperature of 25 °C and fed ad libitum. $Scn1a^{Stop/+}$ knock-in mice were maintained in a Sv129 background and crossed with CMV-Cre (C57BL/6 N) to generate the $Scn1a^{Rec/+}$ mice, with C57BL/6 J to generate F1 mice for the study, and with GAD67-GFP (C57BL/6 N) mice for patch-clamp analysis. B6.Cg-Gt(ROSA)26Sor$^{tm14(CAG-tdTomato)Hze}$/J (Ai14) are from Jackson (# 007914). All procedures were performed according to protocols approved by the internal IACUC and reported to the Italian Ministry of Health according to the European Communities Council Directive 2010/63/EU.

**Generation of $Scn1a^{Stop/+}$ knock-in mice**. The gene targeting vector was built by cloning a genomic fragment of 8.1 kb and inserting the floxed STOP-cassette (modified from Jackson et al. 2011)[52] into ScaI restriction site at 5.6 kb from the 5' end 2.5 Kb from the 3' end of the 8 kb fragment. After linearization, it was electroporated into E14 129Ola ES cells previously assessed for germline competence[53]. 103 ES clones were isolated, expanded and tested for correct genomic integration by southern blot on genomic DNA digested with EcoRI or EcoRV or SphI restriction enzymes and hybridized with a 800 bp probe targeting the 5' arm and 700 bp probe targeting the 3'arm. 6/103 clones resulted to be heterozygous for floxed STOP cassette insertion in Scn1a gene locus. One of them (clone 3A5) was injected into C57Bl/6 blastocysts and transferred to pseudo-pregnant females by the personnel of the Institutional Mouse Transgenic Core Facility at OSR. Two chimeras with a degree of chimerism above 70% were obtained and were employed to establish a mouse line by backcrossing with 129 Sv mice. Genotyping PCR (Fig. 1a) was set with the following primers: WT_F: CCAGGTGAGAGCTATAT GAAAGCATGTAGG; WT_R: AGCCCTGGCTGTCTTGAAACTCAC; STOP_R: CCTACATTTTGAATGGAAGGATTGGAGCTACG.

**AAV- PHP.eB preparation**. For AAV production, replication- incompetent, recombinant viral particles were produced in 293 T cells by polyethylenimine (PEI) (Polyscience) co-transfection of three different plasmids: a transgene-containing plasmid, a packaging plasmid for rep and cap genes, and pAdDeltaF6 for the three adenoviral helper genes. The cells and supernatant were harvested at 120 h. Cells were lysed in Tris buffer (50 mM Tris (pH 8.5), and 150 mM NaCl; Sigma-Aldrich) by repetitive freezing-thawing cycles (3 times), lysed in Tris buffer, and combined with correspondent cell lysates. To clarify the lysate, benzonase treatment was performed (250 U/mL, 37 °C for 30 min; Sigma-Aldrich) in the presence of 1 mM MglCl$_2$ (Sigma-Aldrich), and cellular debris was separated by centrifugation (2000 × $g$, 30 min). The viral phase was isolated by an iodixanol step gradient (15%, 25%, 40%, and 60% Optiprep; Sigma-Aldrich) in the 40% fraction and concentrated in PBS with a 100,000 molecular weight cutoff concentrator (Vivaspin 20, Sartorius Stedim). Virus titers were determined using AAVpro Titration Kit Ver2 (TaKaRa).

**AAV-PHP.eB delivery in neonatal and adult mice**. Vascular injection in perinatal pups was performed as previously described[54] by facial vein injection of $0.3 \times 10^{12}$ vg/mouse in a total volume of 30 ul of PBS. Pups were anesthetized directly on wet ice for 30–60 s and after injection they were put on a warm pad for 2–3 min to recover and rewarm. Systemic injection in juvenile and adult mice was performed in a restrainer that positioned the tail in a heated groove. The tail was swabbed with alcohol and then injected intravenously with a $10^{12}$ vg/mouse in a total volume of 100 μl of PBS.

**Extraction of total RNA from brain tissue and qPCR**. RNA extraction was performed using TRIzol$^{TM}$ reagent (ThermoFisher Scientific) according to the manufacturer's instructions. For qRT-PCR, cDNA synthesis was performed using the ImProm-II Reverse Transcription System (Promega). qRT-PCR was performed in duplicate with custom-designed oligos using Titan HotTaq EvaGreen qPCR Mix 5x (BIOATLAS). The fold change value of the expression of the gene of interest was determined with respect to the control. Primers: 18S_F GGTGAAATTCTTGGAC CGGC and 18S_R GACTTTGGTTTCCCGGAAGC; Scn1a_Ex1_F GGTCCTGGT GGTACAAGCACT and Scn1a_Ex1_R GAGGCTGCAGGAAGCTGAG; Scn1a_

Ex7_F CACCAACGCTTCCCTTGAGG and Scn1a_Ex7_R TGGACATTGGCCTG CATCAG.

**Membrane protein extraction and immunoblotting**. Cerebral cortex, hippocampus, striatum and cerebellum from $Scn1a^{+/+}$,$Scn1a^{Stop/+}$ -Ctrl and $Scn1a^{Stop/+}$-Cre mice were dissected and homogenized using the Mem-PER Plus Membrane Protein Extraction Kit (Thermo Fisher Scientific) according to manufacturer's instructions to achieve enrichment of membrane-bound proteins. Protein extracts were quantified using the Pierce BCA Protein Assay Kit (Thermo Fisher Scientific) following manufacturer's instructions. Samples were boiled at 70 °C per 5 min and then analysed by Western blotting. Briefly, proteins were separated according to their molecular weights on the NuPage$^{TM}$ 3–8% Tris-Acetate gradient gels and subsequently were transferred to a nitrocellulose membrane (GE Healthcare). Primary antibodies for Na$_v$1.1 (rabbit, 1:200, Millipore) and Calnexin (rabbit, 1:2000, Sigma-Aldrich) were used and anti-rabbit HRP (1:5000, Dako) was used as secondary antibody. The densitometric analyses of the corresponding protein bands were performed by ImageJ by normalizing Na$_v$1.1 on calnexin signal.

**Viral copy number**. Total DNA was isolated from animal tissues (cortex and liver) using the Qiagen DNeasy Blood & Tissue Kits (QIAGEN) following the manufacturer's instruction. The quantification of viral DNA copies expressing Cre in PHP-e.B transduced mice Cre viral copy number was calculated by qRT-PCR relative to control heterozygous or homozygous on transgenic mice in which one or two copies of Cre for genome were present. Genomic DNA from cerebral cortices and livers of $Scn1a^{Stop/+}$-Ctrl mice were used as negative controls. The DNA levels were normalized against an amplicon from a single-copy mouse gene, $Lmnb2$, amplified from genomic DNA. Lmn2b_F CTGAGGGTTGCAGGCAGT AG Lmn2b_R TGTGGACAGACCTGGGTAGG; Cre_F GATTTACGGCGCTAA GGATGAC

Cre_R TGCATGATCTCCGGTATTGAAAC

**Immunostaining**. Mice were anesthetized and perfused with paraformaldehyde 4%/PBS. Brains were extracted and post-fixed overnight in paraformaldehyde 4%/PBS, then cryoprotected in sucrose 30%/PBS and frozen in isopentane (Sigma). 50 μm-thick coronal brain sections were cut with a cryostat and free-floating sections were used for immunofluorescence analyses as previously shown[6]. Briefly, after a quick wash in PBS, brain sections were blocked in a solution containing 10% donkey serum, 0.3% TritonX-100 in PBS and then incubated with primary antibodies diluted in the same blocking solution. The following primary antibodies were used: anti-Cre recombinase (mouse, 1:1000, Millipore), anti-NeuN (rabbit, 1:1000, Millipore), anti-GFAP (chicken, 1:1000, Abcam), anti-Glutamine synthetase GS (mouse, 1:1000, Millipore), anti-Iba1 (rabbit, 1:1000, Wako), anti-CD68 (rat, 1:1000, Abcam), anti-Vimentin (chicken, 1:1000, Abcam). Pictures were taken with Leica TCS SP8 confocal microscopy or with a Nikon Eclipse 600 fluorescence microscope.

For GFAP + /GS + and Iba1+ cell counting confocal images with a z-step size of 1 μm were taken using a 40x magnification lens and analysed with the cell counter tool using Photoshop CC2015. Vimentin staining was quantified on confocal images with a z-step size of 1 μm by using ImageJ software generating a macro for the automated image analyses. Sholl analyses were performed to determine astrocyte morphology complexity by using the sample neurite tracer ImageJ tool.

**Survival monitoring**. Mice were monitored daily for survival without knowing the genotype and treatment. Survival data were analyzed for each genotype and survival curves were generated using Prism (GraphPad Software, Inc., CA, USA).

**EEG recordings and analysis**. Mice were implanted under 1.5% isoflurane anesthesia with a wireless radiofrequency transmitter (ETA-F10, Data Science International (DSI)) placed subcutaneously in the back. The recording electrodes were implanted bilaterally in the prefrontal cortex (from bregma, mm: AP -1; ML ± 1) and fixed with white glass ionomer cement. EEG signal was wireless transmitted to MX200 (DSI), digitally recorded using Ponemah (DSI), and sampled at a frequency of 500 Hz. For EEG analysis, Neuroscore software 3.39318-1(DSI) was used. For spontaneous seizures detection, EEG traces were first band-pass filtered between 5 and 70 Hz. After artifact exclusion based on signal amplitude, seizures were defined as transient changes (5-700 ms) in EEG traces (>5 SD), using an automated protocol. Video recordings were then visually inspected to confirm seizures.

**Seizure thermic induction**. We adopted a previously published protocol[55] (with some modifications). Briefly, mice were placed in a glass becker and heated with an infrared heat lamp (HL-1, Phisitemp) to gradually increase the body temperature, controlled by a TCAT-2DF thermo controller (Phisitemp). Mouse rectal temperature was continuously controlled by using a RET-4 probe (Phisitemp). Seizures were recognized by EEG recording and video analyses. Mice were recorded at baseline for 15 min, then seizures were evoked by increasing the body temperature by 0.5 °C every 30 s. The heating lamp was the switched-off to allow recovery and

mice were then monitored until the EEG and temperature returned to the baseline or until death occurred.

**Brain slice electrophysiology.** $Scn1a^{Stop/+}$ mice were crossed with GAD67-GFP knock-in mice[22]. After genotyping, $Scn1a^{+/+}$; and $Scn1a^{Stop/+}$:GAD67-GFP$^+$ were tail-vein injected with PHP-e.B-Ctrl or PHP-e.B-Cre viruses. At P45-54, mice were sacrificed after deep isoflurane anesthesia, and brains were extracted. 350-μm-thick coronal hippocampal sections were cut using a Leica VT 1200 vibratome. After the cut, the slices were allowed to recover for 30 min at 32 °C in modified artificial cerebrospinal fluid (ACSF) containing 92 mM sucrose, 87 mM NaCl, 2.5 mM KCl, 1.25 mM NaH$_2$PO$_4$, 25 mM NaHCO$_3$, 25 mM glucose, and 10 mM MgSO$_4$ aerated with 95% O$_2$ and 5% CO$_2$ (pH 7.4); slices were then allowed to recover at room temperature for at least 45 min before recording.

Current-clamp recordings were performed using a MultiClamp 700B amplifier (Molecular Devices) with pCLAMP 10 software. Signals were low-pass-filtered at 10 kHz and sampled at 50–100 kHz; the signal was digitized using a Digidata 1550 D/A converter (Molecular Devices).

Cells were held at 30 °C–32 °C. The extracellular solution contained 125 mM NaCl, 25 mM NaHCO$_3$, 2 mM CaCl$_2$, 2.5 mM KCl, 1.25 mM NaH$_2$PO$_4$, 1 mM MgSO$_4$, and 10 mM D-glucose aerated with 95% O$_2$ and 5% CO$_2$ (pH 7.4). For current-clamp recordings, the internal solution contained the patch pipette contained 124 mM KH$_2$PO$_4$, 5 mM KCl, 2 mM MgCl$_2$, 10 mM NaCl, 10 mM HEPES, 0.5 mM EGTA, 2 mM Na-ATP, and 0.2 mM Na-GTP (pH 7.25, adjusted with KOH). Neurons with unstable resting potential (or more than −50 mV) and/or holding current of more than 200 pA at −70 mV were discarded. Bridge balance compensation was applied.

CA1 Gad67GFP$^+$ hippocampal neurons were classified into fast spiking and not fast spiking interneurons according to their firing profile (AP frequency, >100 Hz at 400 pA current step, fast AHP).) Current step protocols were used to evoke APs, injecting 500 ms-long depolarizing current steps of increasing amplitude (Δ 10 pA). Passive properties were calculated from the hyperpolarizing steps of the current-clamp step protocol. Input resistance is an average of three steps (2 negative and 1 positive) and is defined as the as ΔV/I. Capacitance was determined in the current-clamp hyperpolarizing step as follows. First, the resistance was determined as voltage derivative (dV)/DI (voltage/current), and then the cell time constant (tau) was obtained, fitting the voltage changing between baseline and hyperpolarizing plateau. Capacitance was calculated as tau/resistance. AP amplitude was calculated from the AP threshold, defined as the voltage at which the first derivative (dV/dt) of the AP waveform reached 10 mV/ms, to the absolute value of the AP peak for the first spike obtained at the current threshold (defined as the minimal current injection able to elicit neuronal firing, determined through 10 pA current steps). Maximal rise and decay slope were defined, respectively, as the maximal and minimal value of the first derivative of the AP waveform.

For voltage-clamp recording the internal solution composition was the following: 125 mM CH$_3$O$_3$SCs, 10 mM NaCl, 2 mM MgCl$_2$, 10 mM HEPES, 2 mM Mg-ATP, and 0.2 mM Na-GTP (pH 7.25, adjusted with CsOH). sIPSCs were recorded from CA1 pyramidal neurons of Gad67GFP mice in voltage-clamp; cells were held at +10 mV for sIPSCs. Pipette capacitance and resistance were always compensated.

Post-synaptic currents (sIPSCs) were analyzed off-line using MiniAnalysis (Synaptosoft). 120 s recording were used to quantify IPSCs frequency and amplitude. First traces were lowpass filtered (8-pole Butterworth) at 800 Hz. PSCs threshold for detection was set to 5 times the baseline noise (1.5/2 pA), usually around 8 to 10 pA; area threshold was set to 3 pA*ms$^{-1}$; baseline was determined as the mean of the 2 ms preceding the IPSC. Individual currents were inspected to reject spurious events. Since IPSCs amplitude does not follow normal distribution, cell amplitude has been expressed as median value and then mean of the median of each cell was plotted. For each cell, 1000 events were considered for IPSCs distribution amplitude distribution analysis.

**Behavior tests**

*Rotarod.* Mice were assessed on an accelerating rotarod (Ugo-Basile, Stoelting Co.). Revolutions per minute were set at an initial value of 10 with a progressive increase to a maximum of 40 rpm across the 5 min test session. Mice received three trials per day during 5 consecutive days (days 1 to 5). Latency to fall was measured by the rotarod timer.

*Open field and novelty.* The test was performed in an open arena (50 × 50 cm). During the open field test mice were allowed to explore the arena for 10 min. Parameters as distance moved (cm), mean speed (cm/s) and thigmotaxis (i.e., the percentage of time spent near the walls) were measured and represented. Subsequently, mice were removed from the open field to place a novel object on the center of the arena. Then, mice were immediately re-entered in the arena and tested for 10 min extra. The time spent sniffing the novel object was measured. Trials were video recorded and automatically analyzed using the video tracking software (Ethovision XT, Noldus).

*Three chambers.* Adult mice were tested in a 60 × 40 cm plexiglass box divided into three chambers; internal walls were provided with an open middle section which allowed free access to each chamber. The test was divided into 2 phases: (A)

habituation: mice could freely explore all three rooms for 10 min, in the lateral rooms two empty wire cages were located; (B) test: after habituation the animals were first confined in the central compartment for 1 min while an unfamiliar mouse was located inside one wire cage. The tested mouse could freely move across the box again for 10 min. Chambers and wire cages were cleaned with ethanol 70% after each mouse. To evaluate social behavior, the time spent sniffing the stranger mouse or the empty cage was manually scored and social index was calculated as (total time sniffing mouse/total time sniffing mouse+empty cage)*100. Time spent in each chamber was also measured. Trials were video recorded using the video tracking software (Ethovision XT, Noldus).

*Radial maze.* The eight-arm radial maze consisted of eight identical arms extending radially from an octagonal platform. It was elevated 80 cm above the floor and surrounded by external cues. A cup containing odorless food was placed at the end of each arm. The protocol was divided into distinct phases: Day1 – food deprivation until the animals had arrived at the 80–85% of their initial weight; during the experiment mice had to maintain this weight. Day 2 – training: put the food in half and at the end of each arm. Release the mouse in the center of the arena, it must eat two of the eight pellets placed at the end of each arm. Day 3–12 –test: pellets are placed only at the end of the eight arms. The mouse is released in the center of the arena to calculate (i) the percentage of the incorrect choices on the total entries (ii) the total number of errors and (iii) the total number of entries in all arms. The maze was cleaned with water and 70% ethanol before the next mouse was placed on the apparatus.

*Water maze.* Mice were trained in a circular pool of 150 cm diameter and 50 cm height according to standardized protocols[56]. The wire-mesh platform was 14 × 14 cm. In the hidden-platform version of the water maze, mice had to locate a hidden platform in a fixed position. The test included an acquisition phase (18 trials, 6 per day, intertrial time 30–40 min) followed by a reversal phase during which the platform was moved to the opposite position (12 trials, 6 per day). The first 60 s of trial 19 (first reversal trial) were considered as probe trial. For the analysis the trials were averaged in blocks of two trials. The following measures were calculated to assess acquisition: escape latency, swim speed, time floating, wall-hugging and the percentage of time in the current quadrant goal (excluding episodes of floating). Spatial selectivity during the probe trial was quantified using the following parameters: percentage of time in the trained quadrant, percentage of time in a circular target zone comprising one-eighth of the pool surface and the annulus crossings. The following measures were calculated to assess platform reversal learning: escape latency and the percentage of time in current quadrant goal (excluding episodes of floating). An automated software algorithm was used to classify video-tracked trials according to the predominant swimming strategy. Eight exclusive categories were defined to capture the gradually improving spatial precision and efficiency during the learning process: circling, floating, wall-hugging, random swimming, scanning, chaining, focal searching and direct swims[57]. Each mouse was categorized according to the most frequently shown strategy (e.g., as floater, as wall-hugger etc). The percentage of mice falling into each category was then computed for each experimental group of mice.

**Rna-seq and bioinformatic analysis.** Cerebral cortex and hippocampus from 4–5 month old mice were dissected and total RNA has been extracted with TRI-zol$^{TM}$ reagent (ThermoFisher Scientific). Only high-quality RNA with a RNA Integrity Number (RIN) of 8 or higher was used. Library preparation and sequencing have been performed with the TruSeq RNA Library Preparation Kit v2 kit (Illumina) by Genewiz. FASTQ sequencing reads were adaptor-trimmed and quality-filtered with Trimmomatic[58], prior to mapping to the mm10 mouse reference genome (https://www.gencodegenes.org/mouse/) with STAR[59]. Gene count normalization and differential gene expression have been done with DESeq2[60]. GeneSCF was employed for functional enrichment analysis and ontology[61]. Low-dimensional embedding of high-dimensional data was done by t-Distributed Neighbor Embedding (t-SNE) machine learning algorithm.

**Statistics.** Statistical analysis performed is shown in each figure legend. Deviation from normal distributions was assessed using D'Agostino-Pearson's test, and the *F*-test was used to compare variances between two sample groups. Student's two-tailed *t*-test (parametric) or the Mann-Whitney test (non-parametric) were used as appropriate to compare means and medians. Fisher's exact test was used to analyze contingency tables. To compare two groups at different time points we used one-way ANOVA or two-way repeated measure ANOVA, followed by *post hoc* tests for functional analysis. Statistical analysis was carried out using Prism (GraphPad Software, Inc., CA, USA) and SPSS (IBM SPSS statistics, NY, USA).

**Reporting Summary.** Further information on research design is available in the Nature Research Reporting Summary linked to this article.

## Data availability

Transcriptomic data are available in the NCBI Gene Expression Omnibus repository with the GSE171191 GEO ID (go to). DEGs among different genotypes analyzed are provided

in Supplementary Data 1 while GO analysis in Supplementary Data 2. Source data are provided with this paper.

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

## Acknowledgements

We thank Marzia Indrigo for help with behavioral study and analysis. We also thank Alessandro Sessa for critical reading of the paper. This work was supported by the Associazione Gruppo Famiglie Dravet and Swiss Dravet Syndrome Association, Telethon GGP19249, Italian Ministry of Health (GR-2016-02363972), CARIPLO Foundation (2016-0532) to G.C., Italian Ministry of Instruction, University PRIN2017 # 2017M95WBA and Dravet Syndrome Foundation 2019 to V.B.

## Author contributions

V.N. performed molecular biology and biochemistry experiments and behavioral analysis on mice; S.B. performed video-EEG analysis and patch clamp; A.S. helped with surgery and breeding; L.S. performed tissue immunofluorescence; M.L. performed in vivo injections for viral delivery; S.G. cloned and produced AAV vector; S. Bido helped with surgery and immunofluorescence analysis; L.M. and F.U. performed RNA-seq bioinformatic analysis; P.G.M helped with mouse ES cultures and gene targeting; P. D'Adamo helped with behavioral data analysis; G.L. helped with the design and analysis of video-EEG and patch-clamp; V.B. and G.C. conceived the study, designed the experiments and wrote the paper.

## Competing interests

The authors declare no competing interests.
