## [Peer Review File · Nature Communications]

Reviewers' Comments:

Reviewer #1:

Remarks to the Author:

The authors present a very nice study showing the reversibility of various symptoms in juvenile and adult mice with an SCN1A defect that can be rescued by activation of Cre recombinase representing another model of Dravet syndrome (DS). The results are urgently warranted and have great importance for the field, since they reveal that a therapy correcting the deleterious gene defect of SCN1A in DS can be probably performed also at later disease stages (corresponding to childhood and adulthood in humans with DS) which are realistic time periods for treatment from a clinical point of view.

The paper is very well written and the main message is very well delivered with a wide range of methods including seizures, behavioral assays, transcriptomics and electrophysiology for rescue of the genetic defect at P30 and functional and behavioral rescue later on (the time point of the various investigations is not always indicated in the text, it is P60 for gene expression and P45-P55 for electrophysiology, but not indicated for example for behavioral studies, this should be complemented). The electrophysiological studies reveal some unexpected and not well explainable results, which are not further investigated. While I do not think it is necessary and appropriate for this paper to find out the reasons for these alterations, the author should be much more cautious in their interpretation (see below for details). At P90, the authors only show that seizures are reversible upon late SCN1A gene activation, but they do not analyze any further features. While I think this is okay for a first report, they should be very cautious with any speculations about the mechanisms, since they simply do not know if there are still any defects of interneuron firing remaining at P90 in their diseased mice (*Scn1aSTOP/+*), if they are corrected and so on, in particular in the light that others found a full reversibility of interneuron firing at later stages in another DS model (Favero et al. *J Neurosci* 2018) which is in disagreement with the authors' data (I am not doubting the results of the authors, but just the knowledge of other data should be a warning for any interpretations of unknown alterations in even later time windows). Actually, it would be very nice to know, if the interneuron firing defect, which has been described so consistently in all DS and other SCN1A models so far at younger age, is more pronounced at around P15-P20 in the authors' mouse model. This would be a straight forward and relatively quick additional experiment to do and reveal if there are any compensatory changes at P30 already, also underlining the authors' own speculation that they look at a relatively benign form of the model that survived the critical phase from P18-24 at which most animals die due to SUDEP. So in my view, this additional experiment should be performed.

Further points to be corrected:

Line 277: The same abbreviation is indicated for fast spiking (FS) and non-fast spiking (FS) interneurons, but later on, non-FS is used for the latter (from line 285 on).

Lines 287 and 292: *Scn1a+/+* should be written in italics.

Line 290: Not sure the term 'non-significant trend' is appropriate. First, 'significant' is rather used in the scientific literature, second, a trend is always non-significant by definition.

Line 304: Should read 'IPSC frequency' or better the 'frequency of IPSCs'.

Line 329: Should read 'regulation of synapses'.

Line 368: Should read 'generalized tonic-clonic seizures'.

Line 403: Should read 'neurodevelopmental'.

Lines 403-405: This first claim sentence is unnecessary and not entirely right, since a similar rescue (although earlier and not in adulthood) has been reported already in 2015 in an animal model of KCNQ2-encephalopathy (Marguet et al. *Nat Med* 2015). Furthermore, the feasibility of rescuing epileptogenic changes has been shown in a focal epilepsy model much earlier (Wykes et al. *Sci Transl Med* 2012).

Line 416: Citations 45 and 46 should be separated by a comma.

Line 460 and following: The authors did not observe a significant increase in excitability of FS interneurons according to the results part (only in non-FS interneurons), so this passage should be corrected. The speculation that Nav1.1 expression increased above normal levels after re-expression at P30 is not supported by any data and not plausible, should be removed. The whole discussion on synaptic activity is not based on solid ground, since interneuron excitability is only partially significantly enhanced. The significance of the observed alterations in IPSCs is restricted to a small change in amplitude, for which no plausible explanation exists, so all this interpretation

should be much more cautious. Basically, the authors do not know what is happening in these mice at later stages of development and should avoid any potentially misleading speculations. Line 474: Should read 'underlie'.

Figure 6: The authors should explain what they mean by current threshold (also in the corresponding main text).

Reviewer #2:

Remarks to the Author:

The manuscript by Valassina et al. presents a new model to study restoration of Scn1a activity in Dravet syndrome (DS) mice. The authors performed a comprehensive analysis from gene expression to behaviour to confirm that Scn1a re-expression could restore the phenotype of DS mice. The paper is well-written, and the data are certainly interesting and important for translational studies of Scn1a restoration in human patients. In particular, the data in Figures 3 and 8 are impressive. However, before accepting for publication, a number of major (and minor) issues should be addressed.

Major points

1) Some of the effects of Scn1a +/- should be long term, a number of cellular characteristics (including connectivity) that have been damaged cannot be restored unless there are some regenerative mechanisms triggered by Scn1a re-expression. Such mechanisms were not reported by RNA-seq analysis, and in fact the reported changes in gene expression between Scn1a +/+ and Scn1a stop/+ Cre are minor (although the time-point for RNA-seq was not perfect). Furthermore, the evidence for long-term non-repairable changes comes from the data in the manuscript, e.g. increased excitability of interneurons upon restoration of Scn1a expression. Thus, the authors should analyze changes in functional connectivity in Scn1a +/+ , Scn1a stop/+ and Scn1a restored mice to provide a clear picture what circuits are restorable and what are not.

2) A large assumption has been made that the symptoms depend mainly on PV INs and that is why P30 is chosen as a developmental time-point of restoration. However, this assumption is not justified, since most studies simply analyzed only PV INs and studies of other IN families, such as SST and VIP, are scarce. Nevertheless, those few studies that did analyze non-PV INs show large and functionally relevant changes in non-PV INs in DS mice, see e.g. De Stasi et al. 2016, Goff et al. 2019. Thus, I would remove such assumption that PV INs are the major drivers of the seizure phenotype and restoration is based on the period of PV maturation. This also has further implications for conclusions, since these are based on the rescue after major PV impairment. However, if other INs (or other neurons in general) have major and earlier contribution to DS phenotype, the conclusions will change. Knocking out and restoration of Scn1a specifically in PV INs with some functional readout will prove PV-based hypothesis of the authors.

3) and related to 2) The authors claim that their electrophysiological experiments were used to distinguish among contribution of PV and non-PV IN dysfunction to the phenotype of DS mice. Studying just CA1 region with rather basic electrophysiology does not really address this. Furthermore, non-FS neuron subtypes are very heterogeneous in the hippocampus. Thus, pooling non-FS together will mask some differences between individual families or subtypes of neurons. A comprehensive ephys study across the cortex and the hippocampus will be necessary to reveal the impact of Scn1a restoration on IN subtypes/families, which could be done e.g. by Patch-seq of GAD-EGFP cells. Otherwise, the authors could confirm that PV INs drive seizure phenotype by specific knockout proposed in 2).

4) Claim that A2 astrocyte markers are consistently upregulated in Scn1a stop/+ has no support. It is probably 1 mouse out of 4 that has some upregulation. The claim should be dropped, or proper statistics should be done to confirm this claim (and it is highly likely that stats will not be significant given that only 4 mice per condition were analyzed for RNA-seq).

Minor points

1) IPSC traces in Fig5k,k' are not the best examples for the quantifications in Fig5m and should be substituted for the traces that indeed represent the quantifications in Fig5m. In addition, it looks

like that some of the mini spikes were not quantified for Scn1a stop/+. The description of sIPSC analysis is too short in the Method part and further details should be provided.

2) What is cell type-specific restoration of Scn1a, in neurons and glia? And a related question, how specific is the restoration of expression – % of those cells that should express Scn1a in wt mice and restore Scn1a expression in Scn1a stop/+ Cre, in other words, what is the efficiency of restoration of the native expression?

3) There is a 2-fold increase in Scn1a mRNA when measured with Exon7 primers, does it mean that the levels of Scn1a mRNA are increased in general? Why Exon 7 is different from Exon 1?

4) Why male mice were selected for behaviour?

5) GO terms should be reported in a Suppl Table with additional details that cannot fit the figure panel, such as genes from GO terms, number of genes found out of total GO term, exact p value etc.

6) Correct misprints, line 315

Correct Figure number, line 306

Correct Figure number for RNAseq data – it is Suppl 7, not 8

Reviewer #3:

Remarks to the Author:

Dravet syndrome is recognized as an epileptic encephalopathy (ILEA). Based on this classification, it is assumed that the aggressive epileptic activity during the maturation of the brain significantly contributes to the progressive deterioration of the overall brain function, including cognitive function, language, and behavior in DS patients. This concept has led to the idea that early and aggressive intervention is key for optimal therapeutic benefit in this illness because some of the later symptoms may be irreversible.

In this manuscript, Valassina and colleagues provide a comprehensive examination of whether reversing the genetic abnormality, the primary source of the disease, once key symptoms have settled in could still provide substantial correction of symptomatology in a novel mouse model of Dravet syndrome.

This new ingenious model was generated by the group; it carries a floxed-stop element between exon 6 and exon 7. They provided strong data showing that in absence of Cre recombinase, this mouse exhibited key phenotypic traits of Dravet syndrome. These symptoms were also reminiscent of those observed in other well-established mouse models of the disease and correlated with decrease Nav 1.1 protein expression in the animal brain. Following the validation of the model, they demonstrated that reinstatement of normal Scn1a activity during embryonic age with a constitutive CMV-Cre prevented the development of symptoms. More strikingly, however, delayed restoration of proper Scn1a activity after the presentation of severe disease symptoms, using a BBB penetrating Cre-virus, was just as effective in reversing the disease as early intervention. In addition, this procedure corrected the cellular and network dysfunctions associated with Dravet syndrome in the model.

This is a very important study for the field, it provides an important new understanding of the mechanisms of DS as it suggests that the majority of the disease manifestations may be functional, not irreversible morphological or pathological deterioration (i.e, abnormal connectivity, neurodegeneration) of the brain. In addition, it provides an important estimate of the ultimate therapeutic benefits of optimal gene therapy for the disease. This information is timely, considering the exponential increase in activity in the field to develop gene therapy for this refractory epilepsy. It brings hope that the field is moving in the right direction to find the cure for this intractable and devastating epilepsy.

The data is very well presented and the evidence appears strong. Appropriate statistical analyses were used and the detailed description of the procedures will help with the reproducibility of the work. The model is an exceptional tool that opens up new frontiers for research in further understanding the mechanisms of Dravet syndrome.

I only have minor concerns:

The sentence of the Abstract and Discussion state that data in this study will "accelerate the therapeutic translation of gene therapies".

Since the tools and methods used in this study will not translate to humans, it seems appropriate to rather indicate that this is a proof-of-concept that highlights the exciting therapeutic potential of optimal gene therapy for Dravet.

Line 174: This sentence needs editing. Starting with this will help "Once the viral efficiency for brain targeting was validated, ..."

Fig 8: DS symptoms usually stabilize after P60 in mice. It is unclear why the control group in this experiment shows a continued increase of seizure frequency.

Reviewer #1 (Remarks to the Author):

The authors present a very nice study showing the reversibility of various symptoms in juvenile and adult mice with an SCN1A defect that can be rescued by activation of Cre recombinase representing another model of Dravet syndrome (DS). The results are urgently warranted and have great importance for the field, since they reveal that a therapy correcting the deleterious gene defect of SCN1A in DS can be probably performed also at later disease stages (corresponding to childhood and adulthood in humans with DS) which are realistic time periods for treatment from a clinical point of view.

The paper is very well written and the main message is very well delivered with a wide range of methods including seizures, behavioral assays, transcriptomics and electrophysiology for rescue of the genetic defect at P30 and functional and behavioral rescue later on (the time point of the various investigations is not always indicated in the text, it is P60 for gene expression and P45-P55 for electrophysiology, but not indicated for example for behavioral studies, this should be complemented). The electrophysiological studies reveal some unexpected and not well explainable results, which are not further investigated. While I do not think it is necessary and appropriate for this paper to find out the reasons for these alterations, the author should be much more cautious in their interpretation (see below for details). At P90, the authors only show that seizures are reversible upon late SCN1A gene activation, but they do not analyze any further features. While I think this is okay for a first report, they should be very cautious with any speculations about the mechanisms, since they simply do not know if there are still any defects of interneuron firing remaining at P90 in their diseased mice (Scn1aSTOP/+), if they are corrected and so on, in particular in the light that others found a full reversibility of interneuron firing at later stages in another DS model (Favero et al. J Neurosci 2018) which is in disagreement with the authors' data (I am not doubting the results of the authors, but just the knowledge of other data should be a warning for any interpretations of unknown alterations in even later time windows).

We thank the Reviewer for appreciating our work and for providing suggestions to improve the delivery of its core message.

We revised the whole manuscript making sure that the time point of each experiment is clearly indicated in the text (behavioral studies were performed in 3–4-month-old mice, highlighted in line 205). We also agree in being more cautious in the speculation on the mechanism underlying the rescue of seizures in DS mice after restoration of normal levels of Nav1.1 at P90. In fact, at this time point we did not perform any electrophysiological study, as the Reviewer pointed out. Therefore, the sentence "Although we did not investigate the functionality of interneurons, we expect that the same mechanisms we described on neurons at P30 may underlie also the recovery at P90" in lines 501-502 of the Discussion was removed from the text.

We are aware of the findings published in Favero et al. J Neurosci 2018, however we do not think that our data disagree with those, conversely, they are complementary. In fact, Favero et al focused their study on Parvalbumin interneurons recorded in primary somatosensory cortex of DS mice highlighting that their impairment of action potential generation in the second and third post-natal weeks is transient and observing a subsequent normalization of excitability starting from P35. However, DS animals continue to seize. To test the possibility that interneurons of other brain areas different from the cerebral cortex remains dysfunctional, we focused our analysis on the hippocampus, in particular on CA1, finding that PV interneurons show a decreased firing rate at high stimulation intensity compared to controls. We had alluded to this in line 295-296 of the Result section and better explained in the Discussion in lines 461-468 and 476-477.

Actually, it would be very nice to know, if the interneuron firing defect, which has been described so consistently in all DS and other SCN1A models so far at younger age, is more pronounced at around P15-P20 in the authors' mouse model. This would be a straight forward and relatively quick additional experiment to do and reveal if there are any compensatory changes at P30 already, also underlining the authors' own speculation that they look at a relatively benign form of the model that survived the critical phase from P18-24 at which most animals die due to SUDEP. So in my view, this additional experiment should be performed.

We thank the Reviewer for this suggestion that is improving the section of our work in which functional data on interneuron activity are presented. We performed patch-clamp experiments in our novel DS mouse model at P18-P23 and P28-33 to assess if the interneuron defect is unambiguously detected already at those developmental stages. In line with previously described mouse models of DS, we reported at P18-P23 a strong impairment in interneuron activity that is still evident, although to less extent, at P30, that is the time in which reactivation of Nav1.1 is induced. Those data have now been included in lines 271-291 of the Result section and in Supplementary Figures 7 and 8.

Further points to be corrected:

Line 277: The same abbreviation is indicated for fast spiking (FS) and non-fast spiking (FS) interneurons, but later on, non-FS is used for the latter (from line 285 on).

We corrected this typo in line 275: non-fast spiking are now indicated as non-FS

Lines 287 and 292: Scn1a+/+ should be written in italics.

Amended.

Line 290: Not sure the term 'non-significant trend' is appropriate. First, 'significant' is rather used the scientific literature, second, a trend is always non-significant by definition.

We agree: "non-significant" was removed from the text.

Line 304: Should read 'IPSC frequency' or better the 'frequency of IPSCs'.

Corrected.

Line 329: Should read 'regulation of synapses'.

Amended.

Line 368: Should read 'generalized tonic-clonic seizures'.

Corrected.

Line 403: Should read 'neurodevelopmental'.

Amended.

Lines 403-405: This first claim sentence is unnecessary and not entirely right, since a similar rescue (although earlier and not in adulthood) has been reported already in 2015 in an animal model of KCNQ2-encephalopathy (Marguet et al. Nat Med 2015). Furthermore, the feasibility of rescuing epileptogenic changes has been shown in a focal epilepsy model much earlier (Wykes et al. Sci Transl Med 2012).

We removed the sentence, following the suggestion of the Reviewer.

Line 416: Citations 45 and 46 should be separated by a comma.

Amended.

Line 460 and following: The authors did not observe a significant increase in excitability of FS interneurons according to the results part (only in non-FS interneurons), so this passage should be corrected.

At this point there was probably a misunderstanding. In fact, upon Cre mediated reactivation of Nav1.1, we detected a significant increase in excitability also in FS interneurons; specifically at high intensity of stimulation the firing of FS interneurons becomes comparable to control condition. However, we agree that the sentence is not completely clear, and we reformulated it in lines 483-486 as follows: "After re-expression of Nav1.1 in P30 Scn1a^{Stop/+}-Cre mice, we found an increased excitability in interneurons compared to controls: in particular, a recovery of FS interneuron excitability was evident only at high stimulation, while an overall increase of non-FS interneuron excitability was observed".

The speculation that Nav1.1 expression increased above normal levels after re-expression at P30 is not supported by any data and not plausible, should be removed.

The Reviewer is probably referring to the sentence in lines 487-490. However, the homeostatic response that we hypothesized to be induced by early interneuron impairment in DS mice is more likely due to be caused by compensatory upregulation of other channel alpha subunits, such as Nav1.2, 1.3, and 1.6 (as we cited in lines 482-483) and not by Nav1.1 upregulation.

The whole discussion on synaptic activity is not based on solid ground, since interneuron excitability is only partially significantly enhanced. The significance of the observed alterations in IPSCs is restricted to a small change in amplitude, for which no plausible explanation exists, so all this interpretation should be much more cautious. Basically, the authors do not know what is happening in these mice at later stages of development and should avoid any potentially misleading speculations.

We followed the Reviewer's suggestion and we removed the specific comment on synaptic activity

Line 474: Should read 'underlie'.

Amended.

Figure 6: The authors should explain what they mean by current threshold (also in the corresponding main text).

We apologize for this inaccuracy. We have now added the definition of current threshold (before indicated in the methods as Rheobase) in lines 280-281 of the Results and lines 684-685 of the Methods.

Reviewer #2 (Remarks to the Author):

The manuscript by Valassina et al. presents a new model to study restoration of Scn1a activity in Dravet syndrome (DS) mice. The authors performed a comprehensive analysis from gene expression to behaviour to confirm that Scn1a re-expression could restore the phenotype of DS mice. The paper is well-written, and the data are certainly interesting and important for translational studies of Scn1a restoration in human patients. In particular, the data in Figures 3 and 8 are impressive. However, before accepting for publication, a number of major (and minor) issues should be addressed.

Major points

1) Some of the effects of Scn1a +/- should be long term, a number of cellular characteristics (including connectivity) that have been damaged cannot be restored unless there are some regenerative mechanisms triggered by Scn1a re-expression. Such mechanisms were not reported by RNA-seq analysis, and in fact the reported changes in gene expression between Scn1a +/- and Scn1a stop/+ Cre are minor (although the time-point for RNA-seq was not perfect). Furthermore, the evidence for long-term non-repairable changes comes from the data in the manuscript, e.g. increased excitability of interneurons upon restoration of Scn1a expression. Thus, the authors should analyze changes in functional

connectivity in Scn1a +/+ , Scn1a stop/+ and Scn1a restored mice to provide a clear picture what circuits are restorable and what are not.

The Reviewer declares that Scn1a mutant mice suffer from “non-repairable changes” in cell features and neuronal connectivity and circuits. However, this speculation is not well supported by data in the literature. In fact, while several studies from different authors converge in unveiling an impaired excitability of cortical and hippocampal GABAergic interneurons (mainly Parvalbumin, Somatostatin and VIP subtypes) caused by the Scn1a gene loss, very few reports on network and circuit alterations in DS mice have been published (Tai et al., PNAS 2014; Tsai et al., Neurobiology of disease 2015; De Stasi et al., Cerebral Cortex 2016; Yan et al., eNeuro 2021). Indeed, many other functional connections are completely unexplored in DS mice, therefore the request to investigate presumptive alterations in functional connectivity is beyond the scope of this study, whose primary goal is to define whether and to which extent the already known phenotypes of Dravet syndrome mice can be reversed in the consolidated stage of the disease. Understanding which circuits undergo alterations in DS mice and which are restored upon Scn1a restoration would be a very interesting and informative study, that however would require years to be set and completed. Furthermore, in the present study we show that connectivity alterations and relative rescue in the “pyramidal CA1- inhibitory neurons” circuit, the most known and studied in the Dravet literature (Figure 6 k-m).

2) A large assumption has been made that the symptoms depend mainly on PV INs and that is why P30 is chosen as a developmental time-point of restoration. However, this assumption is not justified, since most studies simply analyzed only PV INs and studies of other IN families, such as SST and VIP, are scarce. Nevertheless, those few studies that did analyze non-PV INs show large and functionally relevant changes in non-PV INs in DS mice, see e.g. De Stasi et al. 2016, Goff et al. 2019. Thus, I would remove such assumption that PV INs are the major drivers of the seizure phenotype and restoration is based on the period of PV maturation. This also has further implications for conclusions, since these are based on the rescue after major PV impairment. However, if other INs (or other neurons in general) have major and earlier contribution to DS phenotype, the conclusions will change. Knocking out and restoration of Scn1a specifically in PV INs with some functional readout will prove PV-based hypothesis of the authors.

We think that this comment probably arises from a misunderstanding. In fact, we do not believe and we never claimed that DS symptoms mainly depend on PV interneuron dysfunction and we did not choose postnatal day 30 (P30) for restoration of Nav1,1 on that basis. Conversely, we clearly explained that, considering that symptoms appear between P14 and P21, P30 was chosen as a first time-point for Nav1.1 restoration after symptom onset (lines 157-159 of the Result section). Indeed, we agree with the Reviewer that the PV-based hypothesis is not correct because different interneuron subtype contribute to different phenotypes of DS. Selective knocking down of Scn1a specifically in PV, SST (Rubinstein et al., Brain 2015) and VIP (Goff et al 2020) has already been performed therefore there is no need to repeat these experiments. In light of these clarification, also the suggestion to selectively reactivate Scn1a in PV interneurons sounds not necessary in this context as we do not have any PV based hypothesis. This study would be relevant only if performed together with the Scn1a reactivation in other interneuron subtypes like Somatostatin and VIP in order to define the relative contribution of the different interneuron classes to the disease phenotype. However, we believe that this analysis is well beyond the current scope of this study which, again, is not meant to define the contribution of the different neuronal sub-classes to the disease manifestations.

3) and related to 2) The authors claim that their electrophysiological experiments were used to distinguish among contribution of PV and non-PV IN dysfunction to the phenotype of DS mice. Studying just CA1 region with rather basic electrophysiology does not really address this. Furthermore, non-FS neuron subtypes are very heterogeneous in the hippocampus. Thus, pooling non-FS together will mask some differences between individual families or subtypes of neurons. A comprehensive ephys study across the cortex and the hippocampus will be necessary to reveal the impact of Scn1a restoration on IN subtypes/families, which could be done e.g. by Patch-seq of GAD-EGFP cells. Otherwise, the authors could confirm that PV INs drive seizure phenotype by specific knockout proposed in 2).

We agree that non-FS interneurons may include different neuronal subtype. Forebrain GABAergic interneurons (INs) are subdivided in multiple sub-types (e.g. PV, SST, CCK, NPY, VIP, Calretinin) with further diversity between hippocampal and cortical regions. However, GAD67-GFP reporter labels mainly PV, SST and Calretinin (Tamamaki et al., J Comp Neurology 2003). We also confirmed those data by performing immunofluorescence for different IN markers in Gad67 GFP brain sections (Figure 1). Relative quantifications of each IN subtype marker confirm that the most abundant INs labeled by GAD67-GFP reporter are PV, SST and Calretinin, with marginal contribution of CCK, NPY and VIP. In light of this and considering that we can easily recognize PV FS-INs from the firing pattern, we can conclude that non-FS INs analyzed in this study are mainly SST and Calretinin.

Given that, we think that it is completely out of scope of this work to perform a functional analysis of all the interneuron subtypes in hippocampal and cortical regions in the DS mutant mice and after Scn1a gene reactivation.

Figure 1: (A) Representative images of CA1 from hippocampal sections of $Scn1a^{Stop/+}$ GAD67-GFP mice stained for anti-PV, -SST, -CCK, -NPY and -VIP in association with anti-GFP to confirm the identity of patched interneurons. (B) Quantification of the percentage of GAD67-GFP+ cells co-expressing each of the interneuron markers listed above (PV, SST, CCK, NPY and VIP) over the total number of GAD67-GFP+ cells.

4) Claim that A2 astrocyte markers are consistently upregulated in $Scn1a^{stop/+}$ has no support. It is probably 1 mouse out of 4 that has some upregulation. The claim should be dropped, or proper statistics should be done to confirm this claim (and it is highly likely that stats will not be significant given that only 4 mice per condition were analyzed for RNA-seq).

We thank the Reviewer for this comment. In fact, mistakenly, we did not submit the tables with the RNAseq data in our previous submission and we now included them as Table 1 and Table 2 containing differentially expressed genes (DEGs) and gene ontology, respectively. Also, statistical significance for each DEG is now reported in Table 1. Moreover, we have prepared the following table showing how genes that are considered as Pan-Astrocytic markers and genes characteristic of A1 and A2 astrocytic phenotype (Liddelow et al., Nature 2017, doi:10.1038/nature21029) are listed: we evidenced in yellow those genes in which alterations of expression in $Scn1a^{Stop/+}$ mice versus $Scn1a^{+/+}$ are statistically significant for both the p value and the p value adjusted, while in orange are those for which only the p value is statistically significant; in white are those whose expression is not altered. Although not all the markers of A2 astrocytes are significantly up regulated in $Scn1a^{Stop/+}$ mice, the proportion of those genes that are up regulated in A2 categories is higher than those in A1 group. This is the reason why we think that inflamed astrocytes in Dravet mouse brains have a more evident A2 phenotype. However, considering that not all the genes that characterize A2 astrocytes

Minor points

1) IPSC traces in Fig5k,k' are not the best examples for the quantifications in Fig5m and should be substituted for the traces that indeed represent the quantifications in Fig5m. In addition, it looks like that some of the mini spikes were not quantified for Scn1a stop/+. The description of sIPSC analysis is too short in the Method part and further details should be provided.

We appreciate the observation of the Reviewer and we choose new images for Figure 5k,k' that are more representative of the quantifications provided. Also, we included in the Method section a more detailed description of sIPSC analysis (lines 694-701):

*"Post-synaptic currents (sIPSCs) were analyzed off-line using MiniAnalysis (Synptosoft). 120s recording were used to quantify IPSCs frequency and amplitude. First traces were lowpass filtered (8-pole Butterworth) at 800Hz. PSCs threshold for detection was set to 5 times the baseline noise (1.5/2pA), usually around 8 to 10 pA; area threshold was set to 3 pA*ms⁻¹; baseline was determined as the mean of the 2ms preceding the IPSC. Individual currents were inspected to reject spurious events. Since IPSCs amplitude does not follow normal distribution, cell amplitude has been expressed as median value and then mean of the median of each cell was plotted. For each cell, 1000 events were considered for IPSCs distribution amplitude distribution analysis."*

2) What is cell type-specific restoration of Scn1a, in neurons and glia? And a related question, how specific is the restoration of expression – % of those cells that should express Scn1a in wt mice and restore Scn1a expression in Scn1a stop/+ Cre, in other words, what is the efficiency of restoration of the native expression?

The vector we used to deliver the Cre recombinase in the brain of Scn1a^{Stop/+} mice is of PHP.eB serotype, that is able to transduce both neurons and glial cells (Chan et al., Nature Neurosci 2017, doi: 10.1038/nn.4593). However, we underlie that our strategy will reactivate Scn1a only in cells (neurons and eventually glial cells) in which it is physiologically expressed. In fact, also if the PHP.eB-Cre virus will transduce cells that normally do not express Scn1a, this will not be an issue and it will not cause aberrant and ectopic reactivation of the gene. This because we knocked-in the STOP cassette directly into Scn1a locus and therefore all the regulative elements of Scn1a will still regulate its expression. We would finally add that it is not completely known which exactly are the cell types that express Scn1a and this information can be hardly obtained due to the lack of availability of a specific antibody anti-Nav1.1 working in immunofluorescence on brain tissues. However, we showed by western blot for Nav1.1 that upon Cre delivery the levels of Nav1.1 in specific brain areas are rescued indicating that we are likely targeting all the cells that express Nav1.1.

3) There is a 2-fold increase in Scn1a mRNA when measured with Exon7 primers, does it mean that the levels of Scn1a mRNA are increased in general? Why Exon 7 is different from Exon 1?

We performed qRT-PCR to detect the expression levels of Scn1a and we used two different primer pairs, one in Exon 1 and the other in Exon 7 that is exactly after the STOP cassette. In fact, our initial goal was mainly to assess if the levels of Scn1a in Scn1a^{Stop/+} mice were halved with both primer pairs or only with those in Exon 7 in comparison to control Scn1a^{+/+}. In this way we could distinguish if the STOP cassette was interrupting transcription (in this case we expected mRNA levels unaltered with exon A primers and halved with exon 7 primers) or if it was inducing complete degradation of produced mRNA, through a non-sense mediated decay (NSMD) related mechanism (in this way we expected halving of mRNA levels with both primer pairs). The results of the qPCR indicate that there is a trend toward the halving of mRNA levels with both primer pairs in Scn1a^{Stop/+} mouse brains with respect to control, suggesting that a combination of the two hypothesized events occurs, and a trend toward an increase over the physiological levels in Scn1a^{Stop/+}; Cre cortex. This increase is even significative when detected with primers in exon 7. We think that the difference between the primers for exon A and primers for exon 7 is likely due to the different efficiency of different primer pairs in the amplification process.

However, although the course of mRNA levels is not completely expected in the three genotypes, what is key is the level of protein and we always observed that Nav1.1 protein level is halved in Scn1a^{Stop/+} and it is rescued to the expression levels of control mouse brains in Scn1a^{Stop/+}; Cre.

4) Why male mice were selected for behaviour?

We thank the Reviewer for this observation. We performed a first set of behavior experiments that included also female mice. However, we realized soon that female mice, even of the control genotype Scn1a+/+ were performing very bad compared to males in cognitive tests, particularly in the radial maze. For this reason, we removed females from that round, and we selected only males for the next ones in order to reduce variability. We do not have a clear explanation for that, however we think that it might be related to the mixed genetic background Sv129/C57Bl6 of tested mice, as others pointed out previously (Holmes et al., Genes, Brain and Behavior 2002 <https://doi.org/10.1046/j.1601-1848.2001.00005.>).

5) GO terms should be reported in a Suppl Table with additional details that cannot fit the figure panel, such as genes from GO terms, number of genes found out of total GO term, exact p value etc.

We apologize for not including those data in our first submission. As aforementioned, we have now included two tables: Supplementary Table1, containing all DEGs with reactive P value and P value adjusted, and Supplementary Table 2, containing the GO categories.

6) Correct misprints, line 315

Amended.

Correct Figure number, line 306

Amended.

Correct Figure number for RNAseq data – it is Suppl 7, not 8

Amended.

Reviewer #3 (Remarks to the Author):

Dravet syndrome is recognized as an epileptic encephalopathy (ILEA). Based on this classification, it is assumed that the aggressive epileptic activity during the maturation of the brain significantly contributes to the progressive deterioration of the overall brain function, including cognitive function, language, and behavior in DS patients. This concept has led to the idea that early and aggressive intervention is key for optimal therapeutic benefit in this illness because some of the later symptoms may be irreversible.

In this manuscript, Valassina and colleagues provide a comprehensive examination of whether reversing the genetic abnormality, the primary source of the disease, once key symptoms have settled in could still provide substantial correction of symptomatology in a novel mouse model of Dravet syndrome.

This new ingenious model was generated by the group; it carries a floxed-stop element between exon 6 and exon 7. They provided strong data showing that in absence of Cre recombinase, this mouse exhibited key phenotypic traits of Dravet syndrome. These symptoms were also reminiscent of those observed in other well-established mouse models of the disease and correlated with decrease Nav 1.1 protein expression in the animal brain. Following the validation of the model, they demonstrated that reinstatement of normal Scn1a activity during embryonic age with a constitutive CMV-Cre prevented the development of symptoms. More strikingly, however, delayed restoration of proper Scn1a activity after the presentation of severe disease symptoms, using a BBB penetrating Cre-virus, was just as effective in reversing the disease as early intervention. In addition, this procedure corrected the cellular and network dysfunctions associated with Dravet syndrome in the model.

This is a very important study for the field, it provides an important new understanding of the mechanisms of DS as it suggests that the majority of the disease manifestations may be functional, not irreversible morphological or pathological deterioration (i.e, abnormal connectivity, neurodegeneration) of the brain. In addition, it provides an important estimate of the ultimate therapeutic benefits of optimal gene therapy for the disease. This information is timely, considering the exponential increase in activity in the field to develop gene therapy for this refractory epilepsy. It brings hope that the field is moving in the right direction to find the cure for this intractable and devastating epilepsy.

The data is very well presented and the evidence appears strong. Appropriate statistical analyses were used and the detailed description of the procedures will help with the reproducibility of the work. The model is an exceptional tool that opens up new frontiers for research in further understanding the mechanisms of Dravet syndrome.

We thank the Reviewer for his/her deep appreciation of our work.

I only have minor concerns:

The sentence of the Abstract and Discussion state that data in this study will "accelerate the therapeutic translation of gene therapies".

Since the tools and methods used in this study will not translate to humans, it seems appropriate to rather indicate that this is a proof-of-concept that highlights the exciting therapeutic potential of optimal gene therapy for Dravet.

We modified the sentence in the Abstract (lines 54-55) and in Discussion (lines 523-525).

Line 174: This sentence needs editing. Starting with this will help "Once the viral efficiency for brain targeting was validated, ..."

We corrected the sentence as suggested.

Fig 8: DS symptoms usually stabilize after P60 in mice. It is unclear why the control group in this experiment shows a continued increase of seizure frequency.

We agree with the Reviewer and indeed we think that our data go in this direction. In fact, in the control group there are two mice in which the seizure frequency is constant or slightly decreases after Ctrl virus delivery, while for the other four mice there is a slight increase during the observation time. Nevertheless, there is no statistical significance in the seizure frequency before and after the Ctrl virus injection in that group of animals.

Reviewers' Comments:

Reviewer #1:

Remarks to the Author:

The authors have addressed all concerns, I have no further comments.

Reviewer #2:

Remarks to the Author:

While the authors answered some of my critiques, there are few points that remain unanswered, please see below. I believe these could be addressed within a short time.

1) The authors did not answer satisfactory to this point, and the original comment is the key for their conclusions:

Could the strategy proposed by the authors functionally restore the circuits that would be impaired by DS or some of the changes cannot be restored? I still think that the point is well taken – the authors restore Scn1a expression at P30, which is at the end period of brain maturation and after symptom onset. They report that they are able to rescue seizures, behaviour, and FS firing properties. However, the rescue is not complete, and some cellular properties of rescue mice are different from Scn1a^{+/-} mice. Why does it happen, because some of the changes are non-repairable?

My previous comment about potential non-repairable changes is based on the data from the current manuscript and not on the previous literature, I even explicitly wrote:

“the evidence for long-term non-repairable changes comes from the data in the manuscript, e.g. increased excitability of interneurons upon restoration of Scn1a expression”.

The authors did not address this comment and rather started to argue about data in the literature. In addition, there are many papers showing long-term changes in Scn1a^{+/-} mice. In fact, this is confirmed by the manuscript itself – there is a significant increase in seizure induction at P120, which are 4-month-old mice = long-term. There are other prominent studies of Scn1a^{+/-} showing behavioural changes in several month-old mice, see e.g. Han et al. Nature 2012 and Yu et al. 2006 Nat Neurosci, both studies also show changes in cellular characteristics and the latter EEG in 4-month-old mice = long-term changes in firing of neuronal assemblies.

I understand that the authors do not want to study functional circuitry and propose this for future studies. In this case, they should at least explicitly describe their theory in the Discussion and Introduction. If they propose that all impairments are repaired by inducing expression at P30 (almost mature circuits), or even at P90 (completely mature circuits), then there should be only minor connectivity or network level changes in Scn1a^{+/-} or DS (again, unless some kind of regeneration processes are going on) and most changes should be in intrinsic properties. This is of colossal importance for potential therapy – rescuing only intrinsic properties is much easier than re-wiring the circuits. Therefore, repairable/non-repairable issue should be further addressed in the Discussion, and the authors should provide further description of those phenotypic/cellular changes that are present in rescue mice in comparison to controls, and where these changes potentially derive from.

2) The reader is misguided by selection of FS neurons and thus such misunderstanding could happen. More careful wording for why FS neurons are chosen will help (=lack of hypothesis that changes in FS drive the phenotype). The fact that non-FS group in the current study is quite heterogeneous precludes making robust conclusions for the role of other IN subtypes in DS.

Minor point 1) It looks like there is something wrong with formatting in Figure 6k' – enlarged traces are much smaller for rescue mice

Minor point 2) Since the authors cannot provide data on efficiency on the restoration of Scn1a expression upon viral infection, they should at least show that their Cre expression levels and infection rates are high in both Neurons and Glia in the hippocampus – the original study by Chan et al. 2017 Nat Neurosci do not have such data.

Reviewer #3:

Remarks to the Author:

In this revised version of the manuscript, the authors addressed all my concerns with satisfaction.

Results from additional experiments are well presented and changes made to the text have greatly improved the quality of the manuscript.

I am pleased to recommend this great and important work for publication. It will significantly advance the field.

Response to Reviewer 2

Reviewer #2 (Remarks to the Author):

While the authors answered some of my critiques, there are few points that remain unanswered, please see below. I believe these could be addressed within a short time.

1) The authors did not answer satisfactory to this point, and the original comment is the key for their conclusions: Could the strategy proposed by the authors functionally restore the circuits that would be impaired by DS or some of the changes cannot be restored? I still think that the point is well taken – the authors restore Scn1a expression at P30, which is at the end period of brain maturation and after symptom onset. They report that they are able to rescue seizures, behaviour, and FS firing properties. However, the rescue is not complete, and some cellular properties of rescue mice are different from Scn1a^{+/-} mice. Why does it happen, because some of the changes are non-repairable?

My previous comment about potential non-repairable changes is based on the data from the current manuscript and not on the previous literature, I even explicitly wrote: “the evidence for long-term non-repairable changes comes from the data in the manuscript, e.g. increased excitability of interneurons upon restoration of Scn1a expression”. The authors did not address this comment and rather started to argue about data in the literature. In addition, there are many papers showing long-term changes in Scn1a^{+/-} mice. In fact, this is confirmed by the manuscript itself – there is a significant increase in seizure induction at P120, which are 4-month-old mice = long-term. There are other prominent studies of Scn1a^{+/-} showing behavioural changes in several month-old mice, see e.g. Han et al. Nature 2012 and Yu et al. 2006 Nat Neurosci, both studies also show changes in cellular characteristics and the latter EEG in 4-month-old mice = long-term changes in firing of neuronal assemblies. I understand that the authors do not want to study functional circuitry and propose this for future studies. In this case, they should at least explicitly describe their theory in the Discussion and Introduction. If they propose that all impairments are repaired by inducing expression at P30 (almost mature circuits), or even at P90 (completely mature circuits), then there should be only minor connectivity or network level changes in Scn1a^{+/-} or DS (again, unless some kind of regeneration processes are going on) and most changes should be in intrinsic properties. This is of colossal importance for potential therapy – rescuing only intrinsic properties is much easier than re-wiring the circuits. Therefore, repairable/non-repairable issue should be further addressed in the Discussion, and the authors should provide further description of those phenotypic/cellular changes that are present in rescue mice in comparison to controls, and where these changes potentially derive from.

In line with this comment of the reviewer, we have now better highlighted those issues in the Discussion.

Our patch-clamp analysis performed at different time points during postnatal development in control and Dravet mice confirms the existence of a homeostatic response in DS interneurons, observed also by others (Favero et al.,2018) that induces changes in interneuron intrinsic properties and increases their firing ability. We hypothesized that a rebalance of the different Na_v proteins can likely contribute to the observed functional compensation (lines 480-487).

These changes seem to be not repairable, given that the re-expression of Nav1.1 does not induce a complete rescue, but it further increases non-FS interneuron excitability. In the Discussion we

speculated about the origin of this increased excitability hypothesizing it is due to restoration of $Na_v1.1$ levels above the compensation mediated by other Nav channels.

Therefore, Scn1a re-expression starting at P30 does not fully normalize the interneuron activity, but rather it induces the establishment of an alternative equilibrium compatible with the rescue of seizures and stable acquirement of normal behavior.

This analysis has been extensively elaborated in lines 488-504 of the Discussion.

2) The reader is misguided by selection of FS neurons and thus such misunderstanding could happen. More careful wording for why FS neurons are chosen will help (=lack of hypothesis that changes in FS drive the phenotype). The fact that non-FS group in the current study is quite heterogeneous precludes making robust conclusions for the role of other IN subtypes in DS.

We have now better explained why interneurons and in particular FS have been analyzed according to the Reviewer's suggestion (lines of 269-272 the Results).

Minor point 1) It looks like there is something wrong with formatting in Figure 6k' – enlarged traces are much smaller for rescue mice

We apologize for this inaccuracy which has been amended in the revised version.

Minor point 2) Since the authors cannot provide data on efficiency on the restoration of Scn1a expression upon viral infection, they should at least show that their Cre expression levels and infection rates are high in both Neurons and Glia in the hippocampus – the original study by Chan et al. 2017 Nat Neurosci do not have such data.

Given the Cre antibody we employed does not detect low levels of Cre protein that are anyway sufficient to cut the STOP cassette in Scn1a locus of Scn1a^{STOP} mice and fully reactivate Scn1a gene expression, to answer this question we injected Gt(ROSA)26Sor^{tm14(CAG-tdTomato)Hze/J} (Ai14) with PHP.eB-Cre AAV and therefore we exploited tdTomato reporter activation to assess the viral efficiency in the transduction of neurons and glial cells in the hippocampus. These data have been inserted in Supplementary Figure 3 and added in the text (lines 171-175). We observed that 80% of Neurons were transduced and around 20% of glial cells. We don't think that this lower efficiency of glial transduction is relevant for our experiments. In fact, the analysis of single cell RNA seq data (PMID: 34608310) in the hippocampus, confirms that Scn1a is mainly expressed in neurons and only marginally in glial cells (Figure 1).

Figure 1

Reviewers' Comments:

Reviewer #2:

Remarks to the Author:

The authors answered my critique and I support publication of this manuscript in Nat Com

Reviewer comments and answers_1

Reviewer #1 (Remarks to the Author):

The authors present a very nice study showing the reversibility of various symptoms in juvenile and adult mice with an SCN1A defect that can be rescued by activation of Cre recombinase representing another model of Dravet syndrome (DS). The results are urgently warranted and have great importance for the field, since they reveal that a therapy correcting the deleterious gene defect of SCN1A in DS can be probably performed also at later disease stages (corresponding to childhood and adulthood in humans with DS) which are realistic time periods for treatment from a clinical point of view.

The paper is very well written and the main message is very well delivered with a wide range of methods including seizures, behavioral assays, transcriptomics and electrophysiology for rescue of the genetic defect at P30 and functional and behavioral rescue later on (the time point of the various investigations is not always indicated in the text, it is P60 for gene expression and P45-P55 for electrophysiology, but not indicated for example for behavioral studies, this should be complemented). The electrophysiological studies reveal some unexpected and not well explainable results, which are not further investigated. While I do not think it is necessary and appropriate for this paper to find out the reasons for these alterations, the author should be much more cautious in their interpretation (see below for details). At P90, the authors only show that seizures are reversible upon late SCN1A gene activation, but they do not analyze any further features. While I think this is okay for a first report, they should be very cautious with any speculations about the mechanisms, since they simply do not know if there are still any defects of interneuron firing remaining at P90 in their diseased mice (*Scn1a*STOP/+), if they are corrected and so on, in particular in the light that others found a full reversibility of interneuron firing at later stages in another DS model (Favero et al. *J Neurosci* 2018) which is in disagreement with the authors' data (I am not doubting the results of the authors, but just the knowledge of other data should be a warning for any interpretations of unknown alterations in even later time windows).

We thank the Reviewer for appreciating our work and for providing suggestions to improve the delivery of its core message.

We revised the whole manuscript making sure that the time point of each experiment is clearly indicated in the text

(behavioral studies were performed in 3–4-month-old mice, highlighted in line 205). We also agree in being more cautious in the speculation on the mechanism underlying the rescue of seizures in DS mice after restoration of normal levels of Nav1.1 at P90. In fact, at this time point we did not perform any electrophysiological study, as the Reviewer pointed out. Therefore, the sentence “Although we did not investigate the functionality of interneurons, we expect that the same mechanisms we described on neurons at P30 may underlie also the recovery at P90” in lines 501-502 of the Discussion was removed from the text.

*We are aware of the findings published in Favero et al. *J Neurosci* 2018, however we do not think that our data disagree with those, conversely, they are complementary. In fact, Favero et al focused their study on Parvalbumin interneurons recorded in primary somatosensory cortex of DS mice highlighting that their impairment of action potential generation in the second and third post-natal weeks is transient and observing a subsequent normalization of excitability starting from P35. However, DS animal continue to seize. To test the possibility that interneurons of other brain areas different from the cerebral cortex remains dysfunctional, we focused our analysis on the hippocampus, in particular on CA1, finding that PV interneurons show a decreased firing rate at high stimulation intensity compared to controls. We had alluded to this in line 295-296 of the Result section and better explained in the Discussion in lines 461-468 and 476-477.*

Actually, it would be very nice to know, if the interneuron firing defect, which has been described so consistently in all DS and other SCN1A models so far at younger age, is more pronounced at around P15-P20 in the authors' mouse model. This would be a straight forward and relatively quick additional experiment to do and reveal if there are any compensatory changes at P30 already, also underlining the authors' own speculation that they look at a relatively benign form of the model that survived the critical phase from P18-24 at which most animals die due to SUDEP. So in my view, this additional experiment should be performed.

We thank the Reviewer for this suggestion that is improving the section of our work in which functional data on interneuron activity are presented. We performed patch-clamp experiments in our novel DS mouse model at P18-P23 and P28-33 to assess if the interneuron defect is unambiguously detected already at those developmental stages. In line with previously described mouse models of DS, we reported at P18-P23 a strong impairment in interneuron activity that is still evident, although to less extent, at P30, that is the time in which reactivation of Na_v1.1 is induced. Those data have now been included in lines 271-291 of the Result section and in Supplementary Figures 7 and 8.

Further points to be corrected:

Line 277: The same abbreviation is indicated for fast spiking (FS) and non-fast spiking (FS) interneurons, but later on, non-FS is used for the latter (from line 285 on).

We corrected this typo in line 275: non-fast spiking are now indicated as non-FS

Lines 287 and 292: Scn1a^{+/+} should be written in italics.

Amended.

Line 290: Not sure the term 'non-significative trend' is appropriate. First, 'significant' is rather used the scientific literature, second, a trend is always non-significant by definition.

We agree: "non-significative" was removed from the text.

Line 304: Should read 'IPSC frequency' or better the 'frequency of IPSCs'.

Corrected.

Line 329: Should read 'regulation of synapses'.

Amended.

Line 368: Should read 'generalized tonic-clonic seizures'.

Corrected.

Line 403: Should read 'neurodevelopmental'.

Amended.

Lines 403-405: This first claim sentence is unnecessary and not entirely right, since a similar rescue (although earlier and not in adulthood) has been reported already in 2015 in an animal model of KCNQ2-encephalopathy (Marguet et al. Nat Med 2015). Furthermore, the feasibility of rescuing epileptogenic changes has been shown in a focal epilepsy model much earlier (Wykes et al. Sci Transl Med 2012).

We removed the sentence, following the suggestion of the Reviewer.

Line 416: Citations 45 and 46 should be separated by a comma.

Amended.

Line 460 and following: The authors did not observe a significant increase in excitability of FS interneurons according to the results part (only in non-FS interneurons), so this passage should be corrected.

At this point there was probably a misunderstanding. In fact, upon Cre mediated reactivation of Nav1.1, we detected a significant increase in excitability also in FS interneurons; specifically at high intensity of stimulation the firing of FS interneurons becomes comparable to control condition. However, we agree that the sentence is not completely clear, and we reformulated it in lines 483-486 as follows: "After re-expression of Nav1.1 in P30 Scn1a^{Stop/+}-Cre mice, we found an increased excitability in interneurons compared to controls: in particular, a recovery of FS interneuron excitability was evident only at high stimulation, while an overall increase of non-FS interneuron excitability was observed".

The speculation that Nav1.1 expression increased above normal levels after re-expression at P30 is not supported by any data and not plausible, should be removed.

The Reviewer is probably referring to the sentence in lines 487-490. However, the homeostatic response that we hypothesized to be induced by early interneuron impairment in DS mice is more likely due to be caused by compensatory upregulation of other channel alpha subunits, such as Nav1.2, 1.3, and 1.6 (as we cited in lines 482-483) and not by Nav1.1 upregulation.

The whole discussion on synaptic activity is not based on solid ground, since interneuron excitability is only partially significantly enhanced. The significance of the observed alterations in IPSCs is restricted to a small change in amplitude, for which no plausible explanation exists, so all this interpretation should be much more cautious. Basically, the authors do not know what is happening in these mice at later stages of development and should avoid any potentially misleading speculations.

We followed the Reviewer's suggestion and we removed the specific comment on synaptic activity

Line 474: Should read 'underlie'.

Amended.

Figure 6: The authors should explain what they mean by current threshold (also in the corresponding main text).

We apologize for this inaccuracy. We have now added the definition of current threshold (before indicated in the methods as Rheobase) in lines 280-281 of the Results and lines 684-685 of the Methods.

Reviewer #2 (Remarks to the Author):

The manuscript by Valassina et al. presents a new model to study restoration of Scn1a activity in Dravet syndrome (DS) mice. The authors performed a comprehensive analysis from gene expression to behaviour to confirm that Scn1a re-expression could restore the phenotype of DS mice. The paper is well-written, and the data are certainly interesting and important for translational studies of Scn1a restoration in human patients. In particular, the data in Figures 3 and 8 are impressive. However, before accepting for publication, a number of major (and minor) issues should be addressed.

Major points

1) Some of the effects of Scn1a +/- should be long term, a number of cellular characteristics (including connectivity) that have been damaged cannot be restored unless there are some regenerative mechanisms triggered by Scn1a re-expression. Such mechanisms were not reported by RNA-seq analysis, and in fact the reported changes in gene expression between Scn1a +/- and Scn1a stop/+ Cre are minor (although the time-point for RNA-seq was not perfect). Furthermore, the evidence for long-term non-repairable changes comes from the data in the manuscript, e.g. increased excitability of interneurons upon restoration of Scn1a expression. Thus, the authors should analyze changes in functional connectivity in Scn1a +/- , Scn1a stop/+ and Scn1a restored mice to provide a clear picture what circuits are restorable and what are not.

The Reviewer declares that Scn1a mutant mice suffer from “non-repairable changes” in cell features and neuronal connectivity and circuits. However, this speculation is not well supported by data in the literature. In fact, while several studies from different authors converge in unveiling an impaired excitability of cortical and hippocampal GABAergic interneurons (mainly Parvalbumin, Somatostatin and VIP subtypes) caused by the Scn1a gene loss, very few reports on network and circuit alterations in DS mice have been published (Tai et al., PNAS 2014; Tsai et al., Neurobiology of disease 2015; De Stasi et al., Cerebral Cortex 2016; Yan et al., eNeuro 2021). Indeed, many other functional connections are completely unexplored in DS mice, therefore the request to investigate presumptive alterations in functional connectivity is beyond the scope of this study, whose primary goal is to define whether and to which extent the already known phenotypes of Dravet syndrome mice can be reversed in the consolidated stage of the disease. Understanding which circuits undergo alterations in DS mice and which are restored upon Scn1a restoration would be a very interesting and informative study, that however would require years to be set and completed. Furthermore, in the present study we show that connectivity alterations and relative rescue in the “pyramidal CA1- inhibitory neurons” circuit, the most known and studied in the Dravet literature (Figure 6 k-m).

2) A large assumption has been made that the symptoms depend mainly on PV INs and that is why P30 is chosen as a developmental time-point of restoration. However, this assumption is not justified, since most studies simply analyzed only PV INs and studies of other IN families, such as SST and VIP, are scarce. Nevertheless, those few studies that did analyze non-PV INs show large and functionally relevant changes in non-PV INs in DS mice, see e.g. De Stasi et al. 2016, Goff et al. 2019. Thus, I would remove such assumption that PV INs are the major drivers of the seizure phenotype and restoration is based on the period of PV maturation. This also has further implications for conclusions, since these are based on the rescue after major PV impairment. However, if other INs (or other neurons in general) have major and earlier contribution to DS phenotype, the conclusions will change. Knocking out and restoration of Scn1a specifically in PV INs with some functional readout will prove PV-based hypothesis of the authors.

We think that this comment probably arises from a misunderstanding. In fact, we do not believe and we never claimed that DS symptoms mainly depend on PV interneuron dysfunction and we did not choose postnatal day 30 (P30) for restoration of Nav1.1 on that basis. Conversely, we clearly explained that, considering that symptoms appear between P14 and P21, P30 was chosen as a first time-point for Nav1.1 restoration after symptom onset (lines 157-159 of the Result section). Indeed, we agree with the Reviewer that the PV-based hypothesis is not correct because different interneuron subtype contribute to different phenotypes of DS. Selective knocking down of Scn1a specifically in PV, SST (Rubinstein et al., Brain 2015) and VIP (Goff et al 2020) has already been performed therefore there is no need to repeat these experiments. In light of these clarification, also the suggestion to selectively reactivate Scn1a in PV interneurons sounds not necessary in this

context as we do not have any PV based hypothesis. This study would be relevant only if performed together with the Scn1a reactivation in other interneuron subtypes like Somatostatin and VIP in order to define the relative contribution of the different interneuron classes to the disease phenotype. However, we believe that this analysis is well beyond the current scope of this study which, again, is not meant to define the contribution of the different neuronal sub-classes to the disease manifestations.

3) and related to 2) The authors claim that their electrophysiological experiments were used to distinguish among contribution of PV and non-PV IN dysfunction to the phenotype of DS mice. Studying just CA1 region with rather basic electrophysiology does not really address this. Furthermore, non-FS neuron subtypes are very heterogeneous in the hippocampus. Thus, pooling non-FS together will mask some differences between individual families or subtypes of neurons. A comprehensive ephys study across the cortex and the hippocampus will be necessary to reveal the impact of Scn1a restoration on IN subtypes/families, which could be done e.g. by Patch-seq of GAD-EGFP cells. Otherwise, the authors could confirm that PV INs drive seizure phenotype by specific knockout proposed in 2).

We agree that non-FS interneurons may include different neuronal subtype. Forebrain GABAergic interneurons (INs) are subdivided in multiple sub-types (e.g. PV, SST, CCK, NPY, VIP, Calretinin) with further diversity between hippocampal and cortical regions. However, GAD67-GFP reporter labels mainly PV, SST and Calretinin (Tamamaki et al., J Comp Neurology 2003). We also confirmed those data by performing immunofluorescence for different IN markers in Gad67 GFP brain sections (Figure 1). Relative quantifications of each IN subtype marker confirm that the most abundant INs labeled by GAD67-GFP reporter are PV, SST and Calretinin, with marginal contribution of CCK, NPY and VIP. In light of this and considering that we can easily recognize PV FS-INs from the firing pattern, we can conclude that non-FS INs analyzed in this study are mainly SST and Calretinin.

Given that, we think that it is completely out of scope of this work to perform a functional analysis of all the interneuron subtypes in hippocampal and cortical regions in the DS mutant mice and after Scn1a gene reactivation.

Figure 1: (A) Representative images of CA1 from hippocampal sections of *Scn1a*^{Stop/+} GAD67-GFP mice stained for anti-PV, -SST, -CCK, -NPY and -VIP in association with anti-GFP to confirm the identity of patched interneurons. (B) Quantification of the percentage of GAD67-GFP+ cells co-expressing each of the interneuron markers listed above (PV, SST, CCK, NPY and VIP over the total number of GAD67-GFP+ cells).

4) Claim that A2 astrocyte markers are consistently upregulated in *Scn1a* stop/+ has no support. It is probably 1 mouse out of 4 that has some upregulation. The claim should be dropped, or proper statistics should be done to confirm this claim (and it is highly likely that stats will not be significant given that only 4 mice per condition were analyzed for RNA-seq).

We thank the Reviewer for this comment. In fact, mistakenly, we did not submit the tables with the RNAseq data in our previous submission and we now included them as Table 1 and Table 2 containing differentially expressed genes (DEGs) and gene ontology, respectively. Also, statistical significance for each DEG is now reported in Table 1.

Moreover, we have prepared the following table showing how genes that are considered as Pan-Astrocytic markers and genes characteristic of A1 and A2 astrocytic phenotype (Liddelow et al.,

Nature 2017, doi:10.1038/nature21029) are listed: we evidenced in yellow those genes in which alterations of expression in $Scn1a^{Stop/+}$ mice versus $Scn1a^{+/+}$ are statistically significant for both the p value and the p value adjusted, while in orange are those for which only the p value is statistically significant; in white are those whose expression is not altered. Although not all the markers of A2 astrocytes are significantly up regulated in $Scn1a^{Stop/+}$ mice, the proportion of those genes that are up regulated in A2 categories is higher than those in A1 group. This is the reason why we think that inflamed astrocytes in Dravet mouse brains have a more evident A2 phenotype. However, considering that not all the genes that characterize A2 astrocytes are significantly up-regulated we mitigated our claim in line 357-360 and 374 of the Result and line 516 of the Discussion.

1) IPSC traces in Fig5k,k' are not the best examples for the quantifications in Fig5m and should be substituted for the traces that indeed represent the quantifications in Fig5m. In addition, it looks like that some of the mini spikes were not quantified for Scn1a stop/+. The description of sIPSC analysis is too short in the Method part and further details should be provided.

We appreciate the observation of the Reviewer and we choose new images for Figure 5k,k' that are more representative of the quantifications provided. Also, we included in the Method section a more detailed description of sIPSC analysis (lines 694-701):

*“Post-synaptic currents (sIPSCs) were analyzed off-line using MiniAnalysis (Synaptosoft). 120s recording were used to quantify IPSCs frequency and amplitude. First traces were lowpass filtered (8-pole Butterworth) at 800Hz. PSCs threshold for detection was set to 5 times the baseline noise (1.5/2pA), usually around 8 to 10 pA; area threshold was set to 3 pA*ms⁻¹; baseline was determined as the mean of the 2ms preceding the IPSC. Individual currents were inspected to reject spurious events. Since IPSCs amplitude does not follow normal distribution, cell amplitude has been expressed as median value and then mean of the median of each cell was plotted. For each cell, 1000 events were considered for IPSCs distribution amplitude distribution analysis.”*

2) What is cell type-specific restoration of Scn1a, in neurons and glia? And a related question, how specific is the restoration of expression – % of those cells that should express Scn1a in wt mice and restore Scn1a expression in Scn1a stop/+ Cre, in other words, what is the efficiency of restoration of the native expression?

The vector we used to deliver the Cre recombinase in the brain of Scn1a^{Stop/+} mice is of PHP.eB serotype, that is able to transduce both neurons and glial cells (Chan et al., Nature Neurosci 2017, doi: 10.1038/nn.4593).

However, we underlie that our strategy will reactivate Scn1a only in cells (neurons and eventually glial cells) in which it is physiologically expressed. In fact, also if the PHP.eB-Cre virus will transduce cells that normally do not express Scn1a, this will not be an issue and it will not cause aberrant and ectopic reactivation of the gene. This because we knocked-in the STOP cassette directly into Scn1a locus and therefore all the regulative elements of Scn1a will still regulate its expression. We would finally add that it is not completely known which exactly are the cell types that express Scn1a and this information can be hardly obtained due to the lack of availability of a specific antibody anti-Nav1.1 working in immunofluorescence on brain tissues. However, we showed by western blot for Nav1.1 that upon Cre delivery the levels of Nav1.1 in specific brain areas are rescued indicating that we are likely targeting all the cells that express Nav1.1.

3) There is a 2-fold increase in Scn1a mRNA when measured with Exon7 primers, does it mean that the levels of Scn1a mRNA are increased in general? Why Exon 7 is different from Exon 1?

We performed qRT-PCR to detect the expression levels of Scn1a and we used two different primer pairs, one in Exon 1 and the other in Exon 7 that is exactly after the STOP cassette. In fact, our initial goal was mainly to assess if the levels of Scn1a in Scn1a^{Stop/+} mice were halved with both primer pairs or only with those in Exon 7 in comparison to control Scn1a^{+/+}. In this way we could distinguish if the STOP cassette was interrupting transcription (in this case we expected mRNA levels unaltered with exon A primers and halved with exon 7 primers) or if it was inducing complete degradation of produced mRNA, through a non-sense mediated decay (NSMD) related mechanism (in this way we expected halving of mRNA levels with both primer pairs). The results of the qPCR indicate that there is a trend toward the halving of mRNA levels with both primer pairs in Scn1a^{Stop/+} mouse brains with respect to control, suggesting that a combination of the two hypothesized events occurs, and a trend toward an increase over the physiological levels in Scn1a^{Stop/+}; Cre cortex. This increase is even significant when detected with primers in exon 7. We

think that the difference between the primers for exon A and primers for exon 7 is likely due to the different efficiency of different primer pairs in the amplification process.

However, although the course of mRNA levels is not completely expected in the three genotypes, what is key is the level of protein and we always observed that Nav1.1 protein level is halved in $Scn1a^{Stop/+}$ and it is rescued to the expression levels of control mouse brains in $Scn1a^{Stop/+}; Cre$.

4) Why male mice were selected for behaviour?

*We thank the Reviewer for this observation. We performed a first set of behavior experiments that included also female mice. However, we realized soon that female mice, even of the control genotype $Scn1a^{+/+}$ were performing very bad compared to males in cognitive tests, particularly in the radial maze. For this reason, we removed females from that round, and we selected only males for the next ones in order to reduce variability. We do not have a clear explanation for that, however we think that it might be related to the mixed genetic background $Sv129/C57Bl6$ of tested mice, as others pointed out previously (Holmes et al., *Genes, Brain and Behavior* 2002 <https://doi.org/10.1046/j.1601-1848.2001.00005>).*

5) GO terms should be reported in a Suppl Table with additional details that cannot fit the figure panel, such as genes from GO terms, number of genes found out of total GO term, exact p value etc. *We apologize for not including those data in our first submission. As aforementioned, we have now included two tables: Supplementary Table 1, containing all DEGs with reactive P value and P value adjusted, and Supplementary Table 2, containing the GO categories.*

6) Correct misprints, line 315

Amended.

Correct Figure number, line 306

Amended.

Correct Figure number for RNAseq data – it is Suppl 7, not 8

Amended.

Reviewer #3 (Remarks to the Author):

Dravet syndrome is recognized as an epileptic encephalopathy (ILEA). Based on this classification, it is assumed that the aggressive epileptic activity during the maturation of the brain significantly contributes to the progressive deterioration of the overall brain function, including cognitive function, language, and behavior in DS patients. This concept has led to the idea that early and aggressive intervention is key for optimal therapeutic benefit in this illness because some of the later symptoms may be irreversible.

In this manuscript, Valassina and colleagues provide a comprehensive examination of whether reversing the genetic abnormality, the primary source of the disease, once key symptoms have settled in could still provide substantial correction of symptomatology in a novel mouse model of Dravet syndrome.

This new ingenious model was generated by the group; it carries a floxed-stop element between exon 6 and exon 7. They provided strong data showing that in absence of Cre recombinase, this mouse exhibited key phenotypic traits of Dravet syndrome. These symptoms were also reminiscent of those observed in other well-established mouse models of the disease and correlated with decrease Nav 1.1 protein expression in the animal brain. Following the validation of the model, they demonstrated that reinstatement of normal $Scn1a$ activity during embryonic age with a constitutive CMV-Cre prevented the development of symptoms. More strikingly, however, delayed restoration of proper $Scn1a$ activity after the presentation of severe disease symptoms, using a BBB penetrating

Cre-virus, was just as effective in reversing the disease as early intervention. In addition, this procedure corrected the cellular and network dysfunctions associated with Dravet syndrome in the model.

This is a very important study for the field, it provides an important new understanding of the mechanisms of DS as it suggests that the majority of the disease manifestations may be functional, not irreversible morphological or pathological deterioration (i.e, abnormal connectivity, neurodegeneration) of the brain. In addition, it provides an important estimate of the ultimate therapeutic benefits of optimal gene therapy for the disease. This information is timely, considering the exponential increase in activity in the field to develop gene therapy for this refractory epilepsy. It brings hope that the field is moving in the right direction to find the cure for this intractable and devastating epilepsy.

The data is very well presented and the evidence appears strong. Appropriate statistical analyses were used and the detailed description of the procedures will help with the reproducibility of the work. The model is an exceptional tool that opens up new frontiers for research in further understanding the mechanisms of Dravet syndrome.

We thank the Reviewer for his/her deep appreciation of our work.

I only have minor concerns:

The sentence of the Abstract and Discussion state that data in this study will "accelerate the therapeutic translation of gene therapies".

Since the tools and methods used in this study will not translate to humans, it seems appropriate to rather indicate that this is a proof-of-concept that highlights the exciting therapeutic potential of optimal gene therapy for Dravet.

We modified the sentence in the Abstract (lines 54-55) and in Discussion (lines 523-525).

Line 174: This sentence needs editing. Starting with this will help "Once the viral efficiency for brain targeting was validated, ..."

We corrected the sentence as suggested.

Fig 8: DS symptoms usually stabilize after P60 in mice. It is unclear why the control group in this experiment shows a continued increase of seizure frequency.

We agree with the Reviewer and indeed we think that our data go in this direction. In fact, in the control group there are two mice in which the seizure frequency is constant or slightly decreases after Ctrl virus delivery, while for the other four mice there is a slight increase during the observation time. Nevertheless, there is no statistical significance in the seizure frequency before and after the Ctrl virus injection in that group of animals.

Reviewer comments and answers_2

Reviewer #1 (Remarks to the Author):

The authors have addressed all concerns, I have no further comments.

Reviewer #2 (Remarks to the Author):

While the authors answered some of my critiques, there are few points that remain unanswered, please see below. I believe these could be addressed within a short time.

1) The authors did not answer satisfactory to this point, and the original comment is the key for their conclusions:

Could the strategy proposed by the authors functionally restore the circuits that would be impaired by DS or some of the changes cannot be restored? I still think that the point is well taken – the authors restore Scn1a expression at P30, which is at the end period of brain maturation and after symptom onset. They report that they are able to rescue seizures, behaviour, and FS firing properties. However, the rescue is not complete, and some cellular properties of rescue mice are different from Scn1a^{+/-} mice. Why does it happen, because some of the changes are non-repairable?

My previous comment about potential non-repairable changes is based on the data from the current manuscript and not on the previous literature, I even explicitly wrote:

“the evidence for long-term non-repairable changes comes from the data in the manuscript, e.g. increased excitability of interneurons upon restoration of Scn1a expression”.

The authors did not address this comment and rather started to argue about data in the literature. In addition, there are many papers showing long-term changes in Scn1a^{+/-} mice. In fact, this is confirmed by the manuscript itself – there is a significant increase in seizure induction at P120, which are 4-month-old mice = long-term. There are other prominent studies of Scn1a^{+/-} showing behavioural changes in several month-old mice, see e.g. Han et al. Nature 2012 and Yu et al. 2006 Nat Neurosci, both studies also show changes in cellular characteristics and the latter EEG in 4-month-old mice = long-term changes in firing of neuronal assemblies.

I understand that the authors do not want to study functional circuitry and propose this for future studies. In this case, they should at least explicitly describe their theory in the Discussion and Introduction. If they propose that all impairments are repaired by inducing expression at P30 (almost mature circuits), or even at P90 (completely mature circuits), then there should be only minor connectivity or network level changes in Scn1a^{+/-} or DS (again, unless some kind of regeneration processes are going on) and most changes should be in intrinsic properties. This is of colossal importance for potential therapy – rescuing only intrinsic properties is much easier than re-wiring the circuits. Therefore, repairable/non-repairable issue should be further addressed in the Discussion, and the authors should provide further description of those phenotypic/cellular changes that are present in rescue mice in comparison to controls, and where these changes potentially derive from.

In line with this comment of the reviewer, we have now better highlighted those issues in the Discussion.

Our patch-clamp analysis performed at different time points during postnatal development in control and Dravet mice confirms the existence of a homeostatic response in DS interneurons, observed also by others (Favero et al., 2018) that induces changes in interneuron intrinsic properties and increases their firing ability. We hypothesized that a rebalance of the different Na_v proteins can likely contribute to the observed functional compensation (lines 480-487).

These changes seem to be not repairable, given that the re-expression of Nav1.1 does not induce a complete rescue, but it further increases non-FS interneuron excitability. In the Discussion we speculated about the origin of this increased excitability hypothesizing it is due to restoration of Nav1.1 levels above the compensation mediated by other Nav channels.

Therefore, Scn1a re-expression starting at P30 does not fully normalize the interneuron activity, but rather it induces the establishment of an alternative equilibrium compatible with the rescue of seizures and stable acquirement of normal behavior.

This analysis has been extensively elaborated in lines 488-504 of the Discussion.

2) The reader is misguided by selection of FS neurons and thus such misunderstanding could happen. More careful wording for why FS neurons are chosen will help (=lack of hypothesis that changes in FS drive the phenotype). The fact that non-FS group in the current study is quite heterogeneous precludes making robust conclusions for the role of other IN subtypes in DS.

We have now better explained why interneurons and in particular FS have been analyzed according to the Reviewer's suggestion (lines of 269-272 the Results).

Minor point 1) It looks like there is something wrong with formatting in Figure 6k' – enlarged traces are much smaller for rescue mice

We apologize for this inaccuracy which has been amended in the revised version.

Minor point 2) Since the authors cannot provide data on efficiency on the restoration of Scn1a expression upon viral infection, they should at least show that their Cre expression levels and infection rates are high in both Neurons and Glia in the hippocampus – the original study by Chan et al. 2017 Nat Neurosci do not have such data.

Given the Cre antibody we employed does not detect low levels of Cre protein that are anyway sufficient to cut the STOP cassette in Scn1a locus of Scn1a^{STOP} mice and fully reactivate Scn1a gene expression, to answer this question we injected Gt(ROSA)26Sor^{tm14(CAG-tdTomato)Hze/J} (Ai14) with PHP.eB-Cre AAV and therefore we exploited tdTomato reporter activation to assess the viral efficiency in the transduction of neurons and glial cells in the hippocampus. These data have been inserted in Supplementary Figure 3 and added in the text (lines 171-175). We observed that 80% of Neurons were transduced and around 20% of glial cells. We don't think that this lower efficiency of glial transduction is relevant for our experiments. In fact, the analysis of single cell RNA seq data (PMID: 34608310) in the hippocampus, confirms that Scn1a is mainly expressed in neurons and only marginally in glial cells (Figure 1).

Figure 1

Reviewer #3 (Remarks to the Author):

In this revised version of the manuscript, the authors addressed all my concerns with satisfaction.

Results from additional experiments are well presented and changes made to the text have greatly improved the quality of the manuscript.

I am pleased to recommend this great and important work for publication. It will significantly advance the field.

Reviewer comments and answers_3

Reviewer #2 (Remarks to the Author):

The authors answered my critique and I support publication of this manuscript in Nat Com